# Fast Samplers for Inverse Problems in Iterative Refinement Models

**Kushagra Pandey**[*]
Department of Computer Science
University of California Irvine
`pandeyk1@uci.edu`

**Ruihan Yang**[*]
Department of Computer Science
University of California Irvine
`ruihan.yang@uci.edu`

**Stephan Mandt**
Department of Computer Science
University of California Irvine
`mandt@uci.edu`

## Abstract

Constructing fast samplers for unconditional diffusion and flow-matching models has received much attention recently; however, existing methods for solving *inverse problems*, such as super-resolution, inpainting, or deblurring, still require hundreds to thousands of iterative steps to obtain high-quality results. We propose a plug-and-play framework for constructing efficient samplers for inverse problems, requiring only *pre-trained* diffusion or flow-matching models. We present *Conditional Conjugate Integrators*, which leverage the specific form of the inverse problem to project the respective conditional diffusion/flow dynamics into a more amenable space for sampling. Our method complements popular posterior approximation methods for solving inverse problems using diffusion/flow models. We evaluate the proposed method's performance on various linear image restoration tasks across multiple datasets, employing diffusion and flow-matching models. Notably, on challenging inverse problems like $4\times$ super-resolution on the ImageNet dataset, our method can generate high-quality samples in as few as *5* conditional sampling steps and outperforms competing baselines requiring 20-1000 steps. Our code will be publicly available at `https://github.com/mandt-lab/c-pigdm`.

## 1 Introduction

Iterative refinement models, such as diffusion generative models and flow matching methods [Sohl-Dickstein et al., 2015, Ho et al., 2020, Song et al., 2020, Lipman et al., 2023, Albergo and Vanden-Eijnden, 2023], have seen increasing popularity in recent months, and much effort has been invested in accelerating unconditional sampling in these models [Pandey et al., 2024, Shaul et al., 2024, Sauer et al., 2024, Karras et al., 2022, Salimans and Ho, 2022, Zhang and Chen, 2023, Lu et al., 2022, Song et al., 2021]. However, while most efficient samplers have been designed in the *unconditional* setup, current methods for solving *inverse* problems, such as deblurring, inpainting, or super-resolution, still require hundreds to thousands of neural network evaluations to achieve the highest perceptual quality. Moreover, in addition to a score function evaluation, a class of existing methods for solving inverse problems using pre-trained unconditional iterative refinement models often involves expensive Jacobian-vector products [Song et al., 2022, Chung et al., 2022a], making a single sampling step quite expensive and therefore, intolerably slow for most practical applications.

---

[*]Equal contribution

38th Conference on Neural Information Processing Systems (NeurIPS 2024).

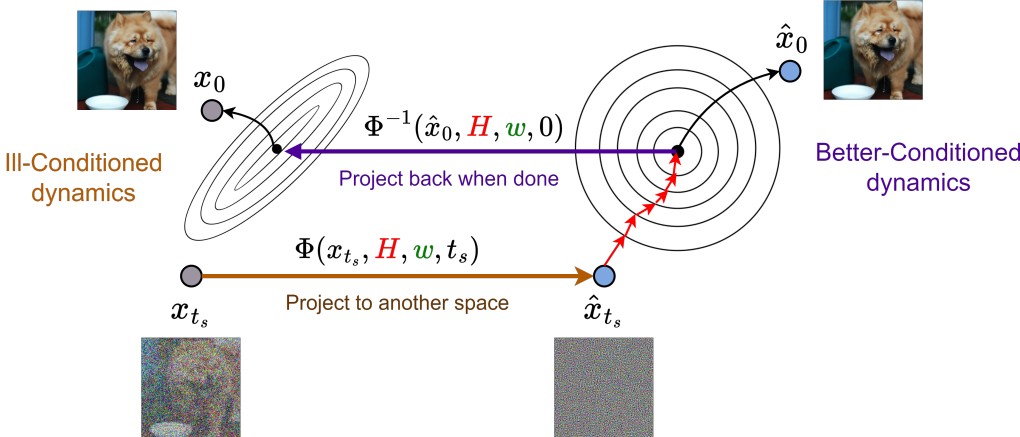

Figure 1: Illustration of Conditional Conjugate Integrators for Fast Sampling in Inverse Problems. Given an initial sampling latent $\mathbf{x}_{t_s}$ at time $t_s$, our sampler projects the diffusion/flow dynamics to a more amenable space for sampling using a projector operator $\Phi$ which is conditioned on the degradation operator $\boldsymbol{H}$ and the sampling guidance scale $w$. The diffusion/flow sampling is then performed in the projected space. Post completion, the generated sample in the projected space is transformed back into the original space using the inverse of the projection operator, yielding the final generated sample. We define the form of the operator $\Phi$ in Section 2.2. Conditional Conjugate Integrators can significantly speed up sampling in challenging inverse problems and can generate high-quality samples in as few as 5 NFEs as compared to existing baselines, which require from 20-1000 NFEs (see Section 3).

This paper presents a principled framework for designing efficient samplers for guided sampling in iterative refinement models, accelerating existing samplers like ΠGDM by an order of magnitude. We present our framework for inverse problems where the degradation operator is known and might be corrupted with additional noise. Crucially, our transformations do not require any re-training and merely rely on some algebraic manipulations of the equations to be simulated.

Intuitively, we expand on the concept of Conjugate Integrators [Pandey et al., 2024] by projecting the conditional generation process in inverse problems to another space that might be better conditioned for faster sampling (See Figure 1). To this end, we separate the linear and non-linear components in the generation process and parameterize the transformation by analytically solving the linear coefficients. By the end of the sampling procedure, we map back to the original sampling space, leading to the concept of *Conditional Conjugate Integrators* that apply to various iterative refinement models such as diffusion models, flows, and interpolants.

In more detail, our main contributions are as follows.

- **Conditional Conjugate Integrators:** We repurpose the recently proposed Conjugate Integrator framework [Pandey et al., 2024] for fast guided sampling in iterative refinement models (diffusions and flows) for linear inverse problems and refer to it as *Conditional Conjugate Integrators*. Next, we design a specific parameterization of the proposed framework, which encodes the structure of the linear inverse problem in the sampler design itself.

- **Theoretical Analysis:** Our parameterization exhibits theoretical properties that help us identify key parameters for sampler design. More specifically, we show that our parameterization (by design) enables recovering high-frequency details early on during sampling. This further enables fast-guided sampling while maintaining good sample quality in the context of inverse problems.

- **Empirical Results**. Empirically, we show that our proposed sampler significantly improves over baselines in terms of sampling efficiency on challenging benchmarks across inverse problems like super-resolution, inpainting, and Gaussian deblurring. For instance, on a challenging 4x superresolution task on the ImageNet dataset, *our proposed sampler achieves better sample quality at 5 steps, compared to 20-1000 steps required by competing baselines.*

Additionally, we extend the proposed framework for noisy and non-linear inverse problems with qualitative demonstrations.

## 2 Fast Samplers for Inverse Problems using Diffusions/Flows.

### 2.1 Background and Problem Statement

Diffusion models define a continuous-time *forward process* (usually with an affine drift) to convert data $\mathbf{x}_0 \in \mathbb{R}^d$ into noise. A learnable *reverse* process is trained to generate data from noise. In this work, we only consider deterministic reverse processes specified as an ODE [Song et al., 2020],

$$d\mathbf{x}_t = \left[ \boldsymbol{F}_t \mathbf{x}_t - \frac{1}{2} \boldsymbol{G}_t \boldsymbol{G}_t^\top \nabla_{\mathbf{x}_t} \log p_t(\mathbf{x}_t) \right] dt. \tag{1}$$

The score is usually intractable and is approximated using a parametric estimator $\boldsymbol{s}_\theta(\mathbf{x}_t, t)$, trained using denoising score matching [Vincent, 2011, Song and Ermon, 2019, Song et al., 2020]. Analogously, one-sided stochastic interpolants [Albergo and Vanden-Eijnden, 2023] define an *interpolant*[2] $\mathbf{x}_t = \alpha_t \mathbf{x}_1 + \gamma_t \mathbf{z}$, where $\mathbf{x}_1 \sim p_{\text{data}}$, and $\mathbf{z} \sim \mathcal{N}(0, \boldsymbol{I})$ to define a transport map between the generative prior (typically an isotropic Gaussian) and the data distribution. Interestingly, the one-sided interpolant induces a vector field $\boldsymbol{b}(\mathbf{x}_t, t) = \mathbb{E}[\dot{\alpha}_t \mathbf{x}_1 + \dot{\gamma}_t \mathbf{z} | \mathbf{x}_t]$, where $\dot{\alpha}_t, \dot{\gamma}_t$ represent the time derivatives of $\alpha_t$ and $\gamma_t$, respectively. The vector field $\boldsymbol{b}(.)$ is typically learned using a neural network approximation $\boldsymbol{b}_\theta(\mathbf{x}_t, t)$. The deterministic interpolant process can then be specified as $d\mathbf{x}_t = \boldsymbol{b}_\theta(\mathbf{x}_t, t) \, dt$. Numerically solving these deterministic generative processes with a sufficient sampling budget can generate plausible samples from noise.

**Problem Statement.** Given a *noisy linear degradation process* (we will consider non-linear processes later) with a degradation operator $\boldsymbol{H}$ specified over an *unobserved* data point $\mathbf{x}_0$,

$$\mathbf{y} = \boldsymbol{H} \mathbf{x}_0 + \sigma_y \mathbf{z}, \quad \mathbf{z} \sim \mathcal{N}(0, \boldsymbol{I}), \ \mathbf{x}_0 \sim p_{\text{data}}, \tag{2}$$

the goal is to recover the original signal $\mathbf{x}_0$. Additionally, given an unconditional pre-trained diffusion or flow matching model, one approach for solving inverse problems is to infer the posterior distribution over the data given the degraded observation, i.e., $p(\mathbf{x}_0|\mathbf{y}) \propto p(\mathbf{y}|\mathbf{x}_0)p(\mathbf{x}_0)$ by simulating the conditional reverse process dynamics i.e.

$$\text{Diffusion:} \qquad d\mathbf{x}_t = \left[ \boldsymbol{F}_t \mathbf{x}_t - \frac{1}{2} \boldsymbol{G}_t \boldsymbol{G}_t^\top \nabla_{\mathbf{x}_t} \log p(\mathbf{x}_t|\boldsymbol{y}) \right] dt, \tag{3}$$

$$\text{Flows:} \qquad d\mathbf{x}_t = \boldsymbol{b}(\mathbf{x}_t, \boldsymbol{y}, t) dt,$$

where $\nabla_{\mathbf{x}_t} \log p(\mathbf{x}_t|\boldsymbol{y})$ and $\mathbf{b}(\mathbf{x}_t, \mathbf{y}, t)$ are the conditional score and velocity estimates, respectively. One approach could be to directly model the conditional score or velocity estimates using a conditional iterative refinement model [Saharia et al., 2022a,b]. However, such approaches are problem-dependent, requiring expensive training pipelines to account for the lack of generalization across inverse problems. Additionally, such methods rely on the availability of paired $(\mathbf{x}_t, \boldsymbol{y})$ measurements, which can be expensive to acquire. Alternatively, *problem-agnostic* methods leverage pre-trained unconditional iterative refinement models to estimate the conditional score or velocity fields and can generalize to different inverse problems without extra training. In this work, we restrict our discussion to the latter and discuss estimating conditional score/velocity fields next.

**Estimating Conditional Score/Velocity from Pretrained Models:** For diffusion models, approximating the conditional score follows directly from Bayes Rule, i.e. $\nabla_{\mathbf{x}_t} \log p(\mathbf{x}_t|\boldsymbol{y}) \approx \boldsymbol{s}_\theta(\mathbf{x}_t, t) + w_t \nabla_{\mathbf{x}_t} \log p(\mathbf{y}|\mathbf{x}_t)$ where $w_t$ is the *guidance weight* (or temperature) of the distribution $p(\mathbf{y}|\mathbf{x}_t)$. Analogously for interpolants (or flows), Pokle et al. [2024] propose the conditional flow dynamics,

$$\boldsymbol{b}(\mathbf{x}_t, \mathbf{y}, t) \approx \boldsymbol{b}_\theta(\mathbf{x}_t, t) + w_t \frac{\gamma_t}{\alpha_t} \left[ \gamma_t \dot{\alpha}_t - \dot{\gamma}_t \alpha_t \right] \nabla_{\mathbf{x}_t} \log p(\mathbf{y}|\mathbf{x}_t). \tag{4}$$

We include a formal proof for the result in Eqn. 4 from an interpolant perspective in Appendix A.1. Since the conditional score and velocity estimates require approximating the term $\nabla_{\mathbf{x}_t} \log p(\mathbf{y}|\mathbf{x}_t)$, we discuss its estimation next.

---

[2]In this work, we use the terms interpolant and flows interchangeably.

**Estimation of the Noise Conditional Score** $\nabla_{\mathbf{x}_t} \log p(\mathbf{y}|\mathbf{x}_t)$**:** The noise conditional distribution $p(\boldsymbol{y}|\mathbf{x}_t)$ can be represented as $p(\boldsymbol{y}|\mathbf{x}_t) = \int p(\boldsymbol{y}|\mathbf{x}_0)p(\mathbf{x}_0|\mathbf{x}_t)d\mathbf{x}_0$. For problem-agnostic models, it is common to approximate the posterior $p(\mathbf{x}_0|\mathbf{x}_t)$ using an unimodal Gaussian distribution [Chung et al., 2022a, Song et al., 2022]. In this work, we restrict our discussion to the posterior approximation in $\Pi$GDM [Song et al., 2022] and its flow variant [Pokle et al., 2024] (named as $\Pi$GFM in our work), $p(\mathbf{x}_0|\mathbf{x}_t) \approx \mathcal{N}(\hat{\mathbf{x}}_0, r_t^2 \boldsymbol{I}_d)$, which yields the following estimate of the conditional score:

$$\nabla_{\mathbf{x}_t} \log p(\mathbf{y}|\mathbf{x}_t) = \frac{\partial \hat{\mathbf{x}}_0}{\partial \mathbf{x}_t}^\top \boldsymbol{H}^\top (r_t^2 \boldsymbol{H}\boldsymbol{H}^\top + \sigma_y^2 \boldsymbol{I}_d)^{-1}(\boldsymbol{y} - \boldsymbol{H}\hat{\mathbf{x}}_0), \tag{5}$$

where $\hat{\mathbf{x}}_0$ is the first-order Tweedie's moment estimate [Stein, 1981]. Our choice of using the $\Pi$GDM approximation is motivated by its expressive posterior approximation $p(x_0|x_t)$ compared to other methods such as DPS or MCG. This makes it an excellent starting point for low-budget sampling.

## 2.2 Conditional Conjugate Integrators

**Conjugate Integrators**   The main idea in conjugate integrators [Pandey et al., 2024] is to project the diffusion dynamics in Eqn. 1 into another space where sampling might be more efficient. The projected diffusion dynamics can then be solved using any numerical ODE solver. On completion, the dynamics can be projected back to the original space to generate samples from the data distribution. To this end, Pandey et al. [2024] introduce an invertible time-dependent affine transformation $\bar{\mathbf{x}}_t = \boldsymbol{A}_t\mathbf{x}_t$. Interestingly, conjugate samplers have theoretical connections to prior work in fast sampling for unconditional diffusion models [Song et al., 2021, Zhang and Chen, 2023, Lu et al., 2022]. We refer the readers to Pandey et al. [2024] for exact details.

### 2.2.1 Conjugate Integrators for Inverse Problems

Next, we design conjugate integrators for linear inverse problems. For simplicity, we discuss noiseless inverse problems, $\sigma_y = 0$, and defer the discussion of noisy inverse problems to Section 2.4. Furthermore, due to space constraints, we present our analysis for diffusion models and defer the discussion of flows to Appendix B. Lastly, without loss of generality, we assume the standard score network parameterization, $\boldsymbol{s}_\theta(\mathbf{x}_t, t) = \boldsymbol{C}_{\text{out}}(t)\boldsymbol{\epsilon}_\theta(\mathbf{x}_t, t)$ where $\boldsymbol{C}_{\text{out}}(t)$ is the notation from the score precondition defined in Karras et al. [2022].

A straightforward way to define conditional conjugate integrators is to treat the score estimate $\nabla_{\mathbf{x}_t} \log p(\mathbf{y}|\mathbf{x}_t)$ as a *black-box* i.e., ignore the structure of the inverse problem. For this case, we formally specify the conjugate integrator formulation as,

**Proposition 1.** (Extended $\Pi$GDM) *For the conditional diffusion dynamics defined in Eqn. 3, introducing a diffeomorphism, $\bar{\mathbf{x}}_t = \boldsymbol{A}_t\mathbf{x}_t$, where,*

$$\boldsymbol{A}_t = \exp\left(\int_0^t \boldsymbol{B}_s - \boldsymbol{F}_s ds\right), \qquad \boldsymbol{\Phi}_t = -\int_0^t \frac{1}{2}\boldsymbol{A}_s\boldsymbol{G}_s\boldsymbol{G}_s^\top \boldsymbol{C}_{out}(s)ds, \tag{6}$$

*induces the following projected diffusion dynamics,*

$$d\hat{\mathbf{x}}_t = \boldsymbol{A}_t\boldsymbol{B}_t\boldsymbol{A}_t^{-1}\hat{\mathbf{x}}_t dt + d\boldsymbol{\Phi}_t\boldsymbol{\epsilon_\theta}(\mathbf{x}_t, t) - \frac{w_t r_t^{-2}}{2}\boldsymbol{A}_t\boldsymbol{G}_t\boldsymbol{G}_t^\top \frac{\partial \hat{\mathbf{x}}_0}{\partial \mathbf{x}_t}^\top (\boldsymbol{H}^\dagger\boldsymbol{y} - \boldsymbol{P}\hat{\mathbf{x}}_0)dt, \tag{7}$$

*where $\boldsymbol{H}^\dagger = \boldsymbol{H}^\top(\boldsymbol{H}\boldsymbol{H}^\top)^{-1}$ and $\boldsymbol{P} = \boldsymbol{H}^\top(\boldsymbol{H}\boldsymbol{H}^\top)^{-1}\boldsymbol{H}$ represent the pseudoinverse and the orthogonal projector operators for the degradation operator $\boldsymbol{H}$. (Proof in Appendix A.2)*

Similar to Pandey et al. [2024], the matrix $\boldsymbol{B}_t$ is a design choice. We refer to the formulation in Eqn. 7 as Extended $\Pi$GDM since for $\boldsymbol{B}_t = 0$, the ODE in Eqn. 7 becomes equivalent to the $\Pi$GDM formulation proposed in Song et al. [2022]. This is because, for $\boldsymbol{B}_t = 0$, Conjugate Integrators are equivalent to DDIM [Song et al., 2021] (See Pandey et al. [2024] for proof). Therefore, the projected diffusion dynamics in Eqn. 7 already present a more generic framework for designing samplers for inverse problems over $\Pi$GDM. In this work, we only explore the parameterization in Eqn. 7 for $\boldsymbol{B}_t = 0$ and hence refer to it simply as $\Pi$*GDM* (analogously $\Pi$*GFM* for flows; see Appendix B).

One characteristic of the formulation in Eqn. 7 is the black-box nature of the conditional score $\nabla_{\mathbf{x}_t} \log p(\mathbf{y}|\mathbf{x}_t)$. However, the inherent linearity in the conditional score can be used to design *better conditioned* (more on this in Section 2.3) conjugate integrators, which we illustrate formally in the form of the following result.

**Proposition 2.** (Conjugate ΠGDM) *Given a noiseless linear inverse problem with $\sigma_y = 0$, a design matrix $\boldsymbol{B} : [0,1] \to \mathbb{R}^{d \times d}$, and the conditional score $\nabla_{\mathbf{x}_t} \log p(\mathbf{y}|\mathbf{x}_t)$ approximated using Eqn. 5, introducing the transformation $\bar{\mathbf{x}}_t = \boldsymbol{A}_t \mathbf{x}_t$, where*

$$\boldsymbol{A}_t = \exp\Big[\int_0^t \boldsymbol{B}_s - \Big(\boldsymbol{F}_s + \frac{w_s r_s^{-2}}{2\mu_s^2}\boldsymbol{G}_s\boldsymbol{G}_s^\top \boldsymbol{P}\Big)ds\Big], \tag{8}$$

*induces the following projected diffusion dynamics:*

$$d\bar{\mathbf{x}}_t = \boldsymbol{A}_t\boldsymbol{B}_t\boldsymbol{A}_t^{-1}\bar{\mathbf{x}}_t dt + d\boldsymbol{\Phi}_y \boldsymbol{y} + d\boldsymbol{\Phi}_s \boldsymbol{\epsilon}_\theta(\mathbf{x}_t, t) + d\boldsymbol{\Phi}_j\Big[\partial_{\mathbf{x}_t}\boldsymbol{\epsilon}_\theta(\mathbf{x}_t, t)(\boldsymbol{H}^\dagger \boldsymbol{y} - \boldsymbol{P}\hat{\mathbf{x}}_0)\Big], \tag{9}$$

*where $\exp(.)$ denotes the matrix exponential, $\boldsymbol{H}^\dagger$, and $\boldsymbol{P}$ are the pseudoinverse and projector operators (as defined previously). Proof in Appendix A.3.*

In this case, the coefficients $\boldsymbol{\Phi}_y$, $\boldsymbol{\Phi}_j$, and $\boldsymbol{\Phi}_s$ depend on time $t$ and the degradation operator $\boldsymbol{H}$ (See Appendix A.3 for full definitions). Intuitively, by including information about the degradation operator $\boldsymbol{H}$ and the guidance scale in the transformation $\boldsymbol{A}_t$ in Eqn. 8, we incorporate the specific structure of the inverse problem in the sampler design, which can have several advantages (more on this in Section 2.3). Moreover, the matrix $\boldsymbol{B}_t$ is a design choice of our parameterization (we will discuss exact choices in Section 2.2.2). We refer to this parameterization as *C-ΠGDM* (analogously *C-ΠGFM for flows; see Appendix B*). In this work, we restrict our discussion to this parameterization and discuss some practical and theoretical aspects next.

### 2.2.2 Practical Design Choices

**Choice of Diffusions and Flows:** While our proposed integrators are applicable to generic diffusion processes [Dockhorn et al., 2022, Pandey and Mandt, 2023] and flows [Ma et al., 2024], we restrict follow-up discussion to VP-SDE [Song et al., 2020] diffusion for which $\boldsymbol{F}_t = -\frac{1}{2}\beta_t \boldsymbol{I}_d, \boldsymbol{G}_t = \sqrt{\beta_t}\boldsymbol{I}_d$ and OT-flows [Liu et al., 20223, Lipman et al., 2023] for which $\alpha_t = t, \gamma_t = 1 - t$. For our score network parameterization, we set $\boldsymbol{C}_{\text{out}}(t) = -1/\sigma_t$, corresponding to the standard $\epsilon$-prediction [Ho et al., 2020, Song et al., 2020] parameterization in diffusion models.

**Choice of $\boldsymbol{B}_t$:** Similar to Pandey et al. [2024], we set $\boldsymbol{B}_t = \lambda \boldsymbol{I}_d$, where $\lambda$ is a time-invariant scalar hyperparameter tuned during inference for optimal sample quality.

**Choice of $w_t$:** Similar to prior work [Song et al., 2022, Pokle et al., 2024], we use an adaptive guidance weight schedule. For diffusion models, we use $w_t = w\mu_t^2 r_t^2$ where $r_t^2 = \frac{\sigma_t^2}{\mu_t^2 + \sigma_t^2}$. Analogously, for flows, we set $w_t = w\alpha_t^2 r_t^2$ where $r_t^2 = \frac{\gamma_t^2}{\alpha_t^2 + \gamma_t^2}$

Having an extra multiplicative factor of $\mu_t^2$ (for VP-SDE) or $\alpha_t^2$ (for flows) stabilizes the numerical computation of coefficients in Eqn. 9 before sampling. We tune the static guidance weight $w$ during inference for optimal sample quality.

**Choice of Start Time:** Given a degradation output $\boldsymbol{y}$, it is common to start diffusion or flow sampling at $\tau < T$ or $\tau > 0$, respectively [Chung et al., 2022b, Song et al., 2022, Pokle et al., 2024]. Consequently, we initialize the diffusion sampling process as $\mathbf{x}_\tau = \mu_\tau \boldsymbol{H}^\dagger \boldsymbol{y} + \sigma_\tau \mathbf{z}$. Analogously for flows, we initialize sampling at $\mathbf{x}_\tau = \alpha_\tau \boldsymbol{H}^\dagger \boldsymbol{y} + \gamma_\tau \mathbf{z}$.

**Choice of the ODE Solver:** Unless specified otherwise, we use the Euler discretization scheme for C-ΠG(D/F)M samplers.

We illustrate a generic C-ΠGDM sampling routine in Algorithm 1 and include additional implementation details in Appendix D. Next, we present some theoretical aspects of our proposed method.

### 2.3 Theoretical Aspects

With the simplifications in Section 2.2.2, the transformation $\boldsymbol{A}_t$ in Eqn. 8 simplifies to:

$$\boldsymbol{A}_t = \exp\Big[\int_0^t \Big(\lambda + \frac{1}{2}\beta_s\Big)ds\boldsymbol{I}_d - \frac{w}{2}\Big(\int_0^t \beta_s ds\Big)\boldsymbol{P}\Big], \tag{10}$$

where $\boldsymbol{P} = \boldsymbol{H}^\top(\boldsymbol{H}\boldsymbol{H}^\top)^{-1}\boldsymbol{H}$ is an orthogonal projection operator.

---

**Algorithm 1** *Conjugate ΠGDM sampling*

---

1: **Input:** Corrupted observation $y$, Corruption operator $\boldsymbol{H}$, Denoiser $\epsilon_{\boldsymbol{\theta}}(.,.)$, Choice of $\boldsymbol{B}_t$, NFE budget $N$, Timestep discretization $\{t_i\}_{i=0}^N$, Diffusion kernel $p(\mathbf{x}_t|\mathbf{x}_0) = \mathcal{N}(\mu_t\mathbf{x}_0, \sigma_t^2\boldsymbol{I}_d)$, Start time $\tau$.
2: **Output:** Clean sample $\hat{\mathbf{x}}_0$

3: Pre-Compute $\{\boldsymbol{A}_{t_i}\}_{i=0}^N$ (Eqn. 8)                ▷ Pre-compute coefficients
4: Pre-Compute $\{\boldsymbol{\Phi}_y^i, \boldsymbol{\Phi}_s^i, \boldsymbol{\Phi}_j^i\}_{i=0}^N$ (see App. A.3)

5: $\mathbf{z} \sim p(\mathbf{x}_T)$              ▷ Draw initial samples from the generative prior
6: $\mathbf{x} = \mu_\tau\boldsymbol{H}^\dagger\boldsymbol{y} + \sigma_\tau\mathbf{z}$      ▷ Initialize using the pseudoinverse (See Chung et al. [2022b])
7: $\bar{\mathbf{x}} = \boldsymbol{A}_\tau\mathbf{x}$                        ▷ Initial Projection Step

8: **for** $n = 0$ **to** $N - 1$ **do**
9:      $h = (t_{n+1} - t_n)$                       ▷ Time step differential
10:      $\mathbf{x} = \boldsymbol{A}_{t_n}^{-1}\bar{\mathbf{x}}$
11:      $\hat{\mathbf{x}}_0 = \frac{1}{\mu_{t_n}}[\mathbf{x} - \sigma_{t_n}\epsilon_{\boldsymbol{\theta}}(\mathbf{x}, t_n)]$             ▷ Tweedie's Estimate
12:      $\boldsymbol{v}_l = h\boldsymbol{A}_{t_n}\boldsymbol{B}_{t_n}\boldsymbol{A}_{t_n}^{-1}\bar{\mathbf{x}} + (\boldsymbol{\Phi}_y^{n+1} - \boldsymbol{\Phi}_y^n)\boldsymbol{y}$       ▷ Linear drift
13:      $\boldsymbol{v}_{nl} = (\boldsymbol{\Phi}_s^{n+1} - \boldsymbol{\Phi}_s^n)\epsilon_{\boldsymbol{\theta}}(\mathbf{x}, t_n) + (\boldsymbol{\Phi}_j^{n+1} - \boldsymbol{\Phi}_j^n)\Big[\partial_\mathbf{x}\epsilon_{\boldsymbol{\theta}}(\mathbf{x}, t_n)(\boldsymbol{H}^\dagger\boldsymbol{y} - \boldsymbol{P}\hat{\mathbf{x}}_0)\Big]$    ▷ Non-Linear drift
14:      $\bar{\mathbf{x}} = \bar{\mathbf{x}} + \boldsymbol{v}_l + \boldsymbol{v}_{nl}$                          ▷ Euler Update
15: **end for**

     **return** $\mathbf{x} = \boldsymbol{A}_{t_N}^{-1}\bar{\mathbf{x}}$            ▷ Project back to original space when done

---

**Computing $\boldsymbol{A}_t$:** While computing the matrix exponential in Eqn. 10 might seem non-trivial, it has several interesting properties that make it tractable to compute. More specifically, the matrix exponential in Eqn. 10 can be simplified as (Proof in Appendix A.4),

$$\boldsymbol{A}_t = \exp(\kappa_1(t))\Big[\boldsymbol{I}_d + (\exp(\kappa_2(t))-1)\boldsymbol{P}\Big], \quad \kappa_1(t) = \int_0^t \Big(\lambda + \frac{1}{2}\beta_s\Big)ds, \quad \kappa_2(t) = -\frac{w}{2}\int_0^t \beta_s ds, \tag{11}$$

where $\exp(.)$ in Eqn. 11 represents the scalar exponential. Furthermore, the integrals in Eqn. 11 are trivial to compute analytically or numerically, making $\boldsymbol{A}_t$ easier to compute. Moreover, $\boldsymbol{A}_t^{-1}$ can also be compactly represented as,

$$\boldsymbol{A}_t^{-1} = \exp(-\kappa_1(t))\Big[\boldsymbol{I}_d + (\exp(-\kappa_2(t)) - 1)\boldsymbol{P}\Big], \tag{12}$$

and is also tractable to compute. Due to the tractability of $\boldsymbol{A}_t$ and $\boldsymbol{A}_t^{-1}$, the projected diffusion dynamics in C-ΠGDM are straightforward to simulate numerically.

**Intuition behind $\boldsymbol{A}_t$:** Next, we analyze several theoretical properties of the transformation matrix $\boldsymbol{A}_t$ in Eqn. 11. More specifically,

$$\bar{\mathbf{x}}_t = \boldsymbol{A}_t\mathbf{x}_t = \exp(\kappa_1(t))\Big[\mathbf{x}_t - (1 - \exp(\kappa_2(t)))\boldsymbol{P}\mathbf{x}_t\Big], \tag{13}$$

Since $\boldsymbol{P} = \boldsymbol{H}^\top(\boldsymbol{H}\boldsymbol{H}^\top)^{-1}\boldsymbol{H}$ is an orthogonal projector, the matrix $\boldsymbol{I}_d - \boldsymbol{P}$ is also an orthogonal projector which projects any vector $\boldsymbol{v}$ in the nullspace of $\boldsymbol{P}$. Therefore, we can decompose the state $\mathbf{x}_t$ into two *orthogonal* components $\mathbf{x}_t = \boldsymbol{P}\mathbf{x}_t + (\boldsymbol{I}_d - \boldsymbol{P})\mathbf{x}_t$. Plugging this form in Eqn. 13,

$$\bar{\mathbf{x}}_t = \exp(\kappa_1(t))\Big[(\boldsymbol{I}_d - \boldsymbol{P})\mathbf{x}_t + \exp(\kappa_2(t))\boldsymbol{P}\mathbf{x}_t\Big], \tag{14}$$

Intuitively, near $t = T$ (i.e., at the start of reverse diffusion sampling), for a large static guidance weight $w$, $\exp(\kappa_2(t)) \to 0$. In this limit, from eqn. 14, $\bar{\mathbf{x}}_t \approx (\boldsymbol{I}_d - \boldsymbol{P})\mathbf{x}_t$. This implies that for a large guidance weight $w$, the diffusion dynamics are projected into the nullspace of the projection operator $\boldsymbol{P}$. Intuitively, for an inverse problem like superresolution, this implies that near the start of the diffusion process, the projected diffusion dynamics correspond to the *denoising of the high-frequency details* missing in $\boldsymbol{P}\mathbf{x}_t$. This is because the projector operation, $\boldsymbol{P}\mathbf{x}_t = \boldsymbol{H}^\dagger\boldsymbol{H}\mathbf{x}_t$ can be interpreted as the pseudoinverse of the noisy degraded state $\mathbf{x}_t$, and, therefore, $(\boldsymbol{I}_d - \boldsymbol{P})\mathbf{x}_t$ represents the high-frequency details missing from the signal component in $\boldsymbol{P}\mathbf{x}_t$.

Moreover, near $t = 0$ (i.e., near the end of reverse diffusion sampling), assuming the guidance weight $w$ is not too large, both coefficients $\exp(\kappa_1(t))$ and $\exp(\kappa_2(t)) \to 1$, which implies $\bar{\mathbf{x}}_t \approx \mathbf{x}_t$. This

implies that near $t = 0$, diffusion happens in the original space, which can prevent over-sharpening artifacts towards the end of sampling. Therefore, we hypothesize that a large $w$ can also lead to over-sharpened results near the end of sampling, resulting in artifacts in the generated samples. Therefore, introducing the projection $A_t$ as defined in Eqn. 10, introduces a tradeoff in the choice of $w$ to control for sample quality. Lastly, since the parameter $\lambda$ controls the *magnitude* of $\bar{x}_t$, it exhibits a similar tradeoff. Indeed, we will empirically demonstrate these tradeoffs in Section 3.3. While our discussion has been limited to diffusion models, a similar theoretical intuition also holds for flows (See Appendix B for proof).

## 2.4 Extension to Noisy and Non-Linear Inverse Problems

While our discussion has been primarily in the context of noiseless linear inverse problems, the conditional Conjugate Integrator framework can also be extended to develop samplers for noisy linear and non-linear inverse problems. We provide a more detailed explanation for the same in App. C.

## 3 Experiments

Next, we empirically demonstrate that our proposed samplers C-ΠGDM/GFM outperform recent baselines on linear image restoration tasks regarding sampling speed vs. quality tradeoff. We then present ablation experiments highlighting the key parameters of our samplers. Lastly, we present design choices for solving noisy and non-linear inverse problems using our proposed framework.

**Models and Dataset:** For diffusion models, we utilize an unconditional pre-trained ImageNet [Deng et al., 2009] checkpoint at 256×256 resolution from OpenAI [Dhariwal and Nichol, 2021][3]. For evaluations on the FFHQ dataset Karras et al. [2019], we use a pre-trained checkpoint from Choi et al. [2021] also at 256×256 resolution. For flow model comparisons, we utilize three publicly available model checkpoints from Liu et al. [20223][4], trained on the AFHQ-Cat [Choi et al., 2020], LSUN-Bedroom Yu et al. [2015], and CelebA-HQ [Karras et al., 2018] datasets. Each flow model was trained at a pixel resolution of $256 \times 256$. For diffusion models, we conduct evaluations on a 1k subset of the evaluation set. For flows, we conduct evaluations on the entire validation set.

**Tasks and Metrics:** We evaluate our samplers qualitatively (see Figure 2) and quantitatively on three challenging linear inverse problems under the noiseless setting. Firstly, we test **Image Super-Resolution**, enhancing images from bicubic-downsampled $64 \times 64$ pixels to $256 \times 256$ pixels. Secondly, we assess **Image Inpainting** performance on images with a fixed free-form center mask. Lastly, we evaluate our samplers on **Gaussian Deblurring**, applying a Gaussian kernel with $\sigma = 3.0$ across a $61 \times 61$ window. We evaluate the performance of each task based on three perceptual metrics: FID [Heusel et al., 2017], KID [Bińkowski et al., 2018] and LPIPS [Zhang et al., 2018].

**Methods and Baselines:** We assess the sample quality of our proposed C-ΠGDM and C-ΠGFM samplers using 5, 10, and 20 sampling steps (denoted as Number of Function Evaluations (NFE)). We conduct an extensive search to optimize the parameters $w$, $\lambda$ and $\tau$ to identify the best-performing configuration based on sample quality. For diffusion baselines, we include DDRM [Kawar et al., 2022], DPS [Chung et al., 2022a], and ΠGDM [Song et al., 2022]. As recommended for DPS [Chung et al., 2022a], we use NFE=1000 for all tasks. For DDRM, we adhere to the original implementation and run it with $\eta_b = 1.0$ and $\eta = 0.85$ at NFE=20. We test our implementation of ΠGDM (see Section 2.2), with NFE values of 5, 10, and 20 and use the recommended guidance schedule of $w_t = r_t^2$ across all tasks. For flow models, we consider the recently proposed method inspired by ΠGDM running on OT-ODE path by Pokle et al. [2024] (which we refer to as ΠGFM; see Appendix B), and similarly run it with NFE values of 5, 10, and 20. We optimize all baselines by conducting an extensive grid search over $w$ and $\tau$ for the best performance (in terms of sample quality).

## 3.1 Quantitative Results

We present the results of our method applied to inverse problems in Table 1, specifically using the CelebA-HQ dataset for flow-based models and the ImageNet dataset for diffusion-based models. For

---

[3]https://github.com/openai/guided-diffusion
[4]https://github.com/gnobitab/RectifiedFlow

| Flow Results | NFE | LPIPS↓ | | KID×10⁻³ ↓ | | FID↓ | |
|---|---|---|---|---|---|---|---|
| | | **C-ΠGFM** | ΠGFM | **C-ΠGFM** | ΠGFM | **C-ΠGFM** | ΠGFM |
| Inpainting | 5 | **0.125** | 0.240 | **17.6** | 167.0 | **26.95** | 161.49 |
| | 10 | **0.074** | 0.188 | **8.0** | 86.6 | **14.64** | 94.91 |
| | 20 | **0.065** | 0.144 | **4.6** | 54.4 | **10.93** | 65.39 |
| Super-Resolution | 5 | **0.063** | 0.091 | **5.5** | 17.5 | **13.08** | 21.84 |
| | 10 | **0.058** | 0.076 | **3.6** | 12.2 | **10.65** | 16.73 |
| | 20 | **0.064** | 0.069 | 3.9 | **3.5** | 11.07 | **10.23** |
| Deblurring | 5 | **0.083** | 0.114 | **3.7** | 10.9 | **12.86** | 18.97 |
| | 10 | **0.077** | 0.088 | **5.0** | 7.0 | **14.41** | 15.09 |
| | 20 | 0.080 | **0.073** | 7.9 | **3.1** | 17.10 | **11.35** |

| Diffusion Results | | **C-ΠGDM** | ΠGDM | DPS | DDRM | **C-ΠGDM** | ΠGDM | DPS | DDRM | **C-ΠGDM** | ΠGDM | DPS | DDRM |
|---|---|---|---|---|---|---|---|---|---|---|---|---|---|
| Super-Resolution | 5 | **0.220** | 0.306 | | | **2.7** | 6.3 | | | **37.31** | 49.06 | | |
| | 10 | **0.206** | 0.252 | 0.252 | 0.318 | **1.6** | 4.8 | 5.8 | 14.1 | **34.22** | 44.30 | 38.18 | 51.64 |
| | 20 | **0.207** | 0.222 | | | **1.7** | 2.5 | | | **34.28** | 37.36 | | |
| Deblurring | 5 | **0.272** | 0.349 | | | **3.89** | 14.1 | | | **44.42** | 63.94 | | |
| | 10 | **0.272** | 0.294 | 0.619 | 0.336 | **3.6** | 5.3 | 59.5 | 12.3 | **43.37** | 47.80 | 139.58 | 62.53 |
| | 20 | 0.268 | **0.259** | | | **3.5** | 4.2 | | | **43.70** | 44.20 | | |

Table 1: Comparison between Conjugate ΠG(D/F)M and other baselines for noiseless linear inverse problems. Top: Flow models (CelebA-HQ) and Bottom: Diffusion Models (ImageNet). Entries in bold show the best performance for a given sampling budget.

a comprehensive review of additional results across different datasets, please refer to Appendix E. Our method consistently surpasses other approaches across all sampling budgets (indicated by NFE) for the inpainting task. Similarly, our flow-based sampler (C-ΠGFM) exhibits superior perceptual quality for image super-resolution at NFEs of 5 and 10. The ΠGFM model only reaches comparable performance at higher NFEs. Remarkably, our diffusion-based sampler C-ΠGDM outperforms all baselines across the entire range of NFEs. Notably, C-ΠGDM outperforms competing baselines requiring 20-1000 NFEs in just 5 sampling steps on the challenging ImageNet dataset, demonstrating a significant speedup in sampling speed while preserving sample quality. A similar pattern is observed in the image deblurring task, where the performance of ΠGDM/ΠGFM approaches that of our method only when the NFE is increased to 20 steps.

Interestingly, we observe a plateau in performance improvements at NFE=20 for both super-resolution and deblurring tasks using our method. This suggests that while our method efficiently utilizes the iterative model under a deterministic path with an Euler solver, further enhancements in performance, particularly at higher NFEs, might require integrating stochastic sampling techniques or more advanced solvers. This potential next step could unlock further gains from our approach in complex image processing tasks.

### 3.2 Qualitative Results

Figure 2 presents a qualitative comparison between our proposed method and the ΠG(D/F)M baseline. The inpainting results in the first column reveal that ΠGFM tends to introduce gray artifacts within the inpainted areas. This issue may stem from the initialization of the parameter $\tau$; optimal performance is achieved when $\tau \geq 0.2$, as established during our parameter tuning phase and corroborated by Pokle et al. [2024]. Consequently, insufficient NFE means ΠGFM cannot effectively eliminate the artifacts associated with the inpainting mask in our experiments. For image super-resolution, our method excels in restoring fine details, particularly evident in high-frequency image components such as human hair and wheat ears. Similarly, for the deblurring task, our method qualitatively outperforms the baseline, especially in mitigating the over-smoothing artifacts (Figure 2, last column). Additional examples are provided in Appendix E.4.

### 3.3 Ablation Studies

In this section, we further explore the impact of the hyperparameters $w$, $\lambda$, and $\tau$, which were identified during our tuning phase and link to the theoretical insights discussed in Section 2.3. We recognize that $\tau$ is particularly task-specific and relatively straightforward to adjust. For instance, tasks such as inpainting require a smaller $\tau$ to prevent masking artifacts, whereas tasks like super-

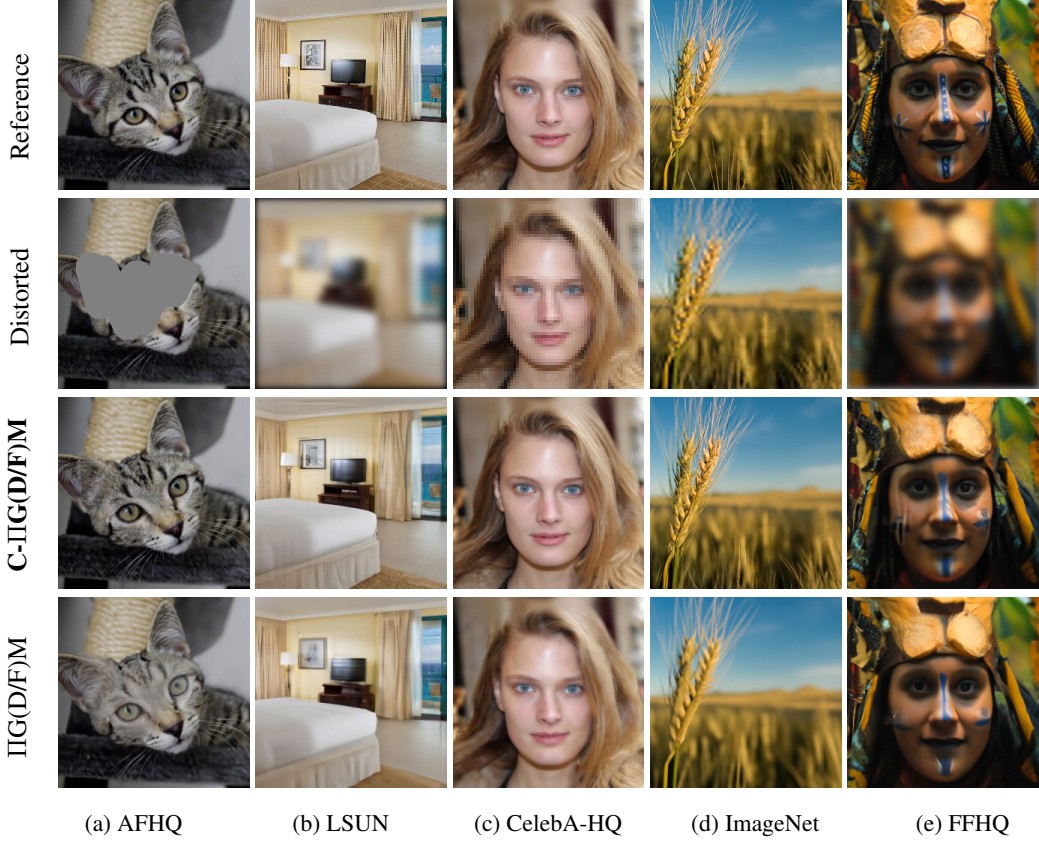

(a) AFHQ      (b) LSUN      (c) CelebA-HQ      (d) ImageNet      (e) FFHQ

Figure 2: Qualitative comparison between C-ΠG(D/F)M and ΠG(D/F)M baselines on five different datasets. (a, b, c) Inpainting, De-blurring, and 4x Super-resolution with C-ΠGFM, respectively. (d,e) 4x Image Super-resolution and De-blurring with C-ΠGDM, respectively. ($\sigma_y = 0$, NFE=5)

resolution or deblurring benefit from a larger $\tau$ to ensure effective initialization. Consequently, our discussion will primarily focus on the effects of $w$ and $\lambda$. Figure 3 illustrates the impact of varying $w$ and $\lambda$ on sample quality for image super-resolution on the CelebA-HQ and ImageNet datasets.

From Figure 3 we make the following observations. Firstly, for both C-ΠGDM and C-ΠGFM samplers, we observe that the optimal value of $\lambda$ can differ from $\lambda = 0$. This illustrates the usefulness of parameterizing $B_t$ in our sampler design. On the contrary, ΠGDM or ΠGFM samplers do not have this flexibility and, therefore, yield sub-optimal sample quality at different sampling budgets. Secondly, we observe that deviating from the optimal $\lambda$ can lead to degradation in sample quality. More specifically, we observed that deviating from our tuned value of $\lambda$ leads to either over-sharpening artifacts or blurry samples (See Figs. 7, 11). This is intuitive since $\lambda$ controls the scale of the transformation $\bar{\mathbf{x}}_t = A_t \mathbf{x}_t$ (see Eqn. 14) and thus plays a significant role in conditioning the projected diffusion dynamics. We observe a similar tradeoff on varying the static guidance weight $w$ where a large magnitude of $w$ can lead to over-sharpened artifacts while a very small guidance weight can lead to blurry samples (See Figs. 6, 10). These empirical observations are consistent with our theoretical analysis in Section 2.3, confirming our theoretical intuition on the role of the sampler parameters $w$ and $\lambda$.

## 4 Related Works

**Fast Unconditional Sampling:** Recent research has significantly advanced the efficiency of the sampling process in unconditional diffusion/flow models [Song et al., 2020, Lipman et al., 2023, Manduchi et al., 2024]. One line of research involves designing efficient diffusion models to improve sampling by design [Karras et al., 2022, Dockhorn et al., 2022, Pandey and Mandt, 2023, Song et al., 2023]. Since our treatment of conditional Conjugate Integrators is quite generic, our method is readily

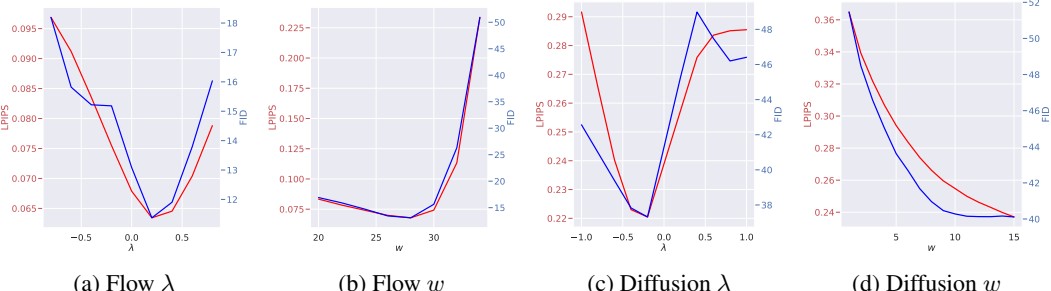

| (a) Flow $\lambda$ | (b) Flow $w$ | (c) Diffusion $\lambda$ | (d) Diffusion $w$ |

Figure 3: Impact of $\lambda$ and $w$ on sampling quality. Red curves and labels represent the LPIPS scores, while blue curves and labels indicate the FID scores.

compatible with most advancements in diffusion model design. Another line of work focuses on distilling a student model from a teacher model, enabling sampling in even a single step [Salimans and Ho, 2022, Meng et al., 2023, Sauer et al., 2024]. However, since these methods require expensive re-training, there has been a significant interest in the development of fast samplers applicable to pretrained diffusion/flow models [Liu et al., 2022, Pandey et al., 2024, Shaul et al., 2024, Zhang and Chen, 2023, Lu et al., 2022, Song et al., 2021, Gonzalez et al., 2023]. Our work falls under the latter line of research, where we develop fast conditional samplers that can be applied to pretrained diffusion models.

**Conditional Iterative Refinement Models** have become prevalent for tasks requiring controlled generation. These models often involve training specialized conditional diffusion models [Saharia et al., 2022b, Yang and Mandt, 2023, Kong et al., 2021, Pandey et al., 2022, Preechakul et al., 2022, Rombach et al., 2022, Podell et al., Ramesh et al., 2022, Peebles and Xie, 2023, Ma et al., 2024, Esser et al., 2024, Chen et al., 2024] and may incorporate classifier-free guidance [Ho and Salimans, 2021] or classifier guidance [Dhariwal and Nichol, 2021, Song et al., 2020] for conditional sampling. These approaches have also spurred research into solving inverse problems related to various image degradation transformations, such as inpainting and super-resolution [Kawar et al., 2022, Chung et al., 2022a, Song et al., 2022, Mardani et al., 2023, Pokle et al., 2024]. Although these methods demonstrate promising outcomes, they are typically bottlenecked by a costly sampling process, emphasizing the need for a fast sampler to address inverse problems efficiently. Recent work Xu et al. [2024] employs a consistency model Song et al. [2023] to enhance posterior approximation, but incorporating an additional model may deviate from our proposal of using a single pre-trained model. DPM-Solver++ [Lu et al., 2023] also tackles the problem of accelerating guided sampling in diffusion models. However, unlike [Lu et al., 2023], we incorporate the structure of the inverse problem in the sampler design.

## 5 Discussion

We present a generic framework for designing samplers for accelerating guided sampling in iterative refinement models. In this work, we explore a specific parameterization of this framework, which incorporates the structure of the inverse problem in sampler design. We provide a theoretical intuition behind our design choices and empirically justify its effectiveness in solving linear inverse problems in as few as 5 sampling steps compared to 20-1000 NFEs required by competing baselines. While our method can serve as an important step toward designing fast-guided samplers, there are several important future directions. Firstly, our parameterization of the transform $A_t$ can be more expressive by learning it directly during the sampling stage. Secondly, in this work, we consider inverse problems with a known degradation operator. Extending our framework for solving blind inverse problems could be an important research direction. Lastly, it would be interesting to adapt our solvers to techniques for solving inverse problems in latent diffusion models [Rout et al., 2024] to enhance sampling efficiency further.

**Broader Impact:** While our work has the potential to make synthetic data generation accessible, the techniques presented in this work should be used responsibly. Moreover, despite good sample quality in a limited sampling budget, restoration can sometimes lead to artifacts in the generated sample which can be undesirable in some domains like medical image analysis.

## Acknowledgments and Disclosure of Funding

The authors thank Justus Will for providing valuable feedback on our manuscript. KP acknowledges support from the HPI Research Center in Machine Learning and Data Science at UC Irvine. SM acknowledges support from the National Science Foundation (NSF) under an NSF CAREER Award IIS-2047418 and IIS-2007719, the NSF LEAP Center, by the Department of Energy under grant DE-SC0022331, the IARPA WRIVA program, the Hasso Plattner Research Center at UCI, and by gifts from Qualcomm and Disney.

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

# Contents

## A   Proofs

### A.1   Proof of Conditional flow dynamics

*Proof.* Given a one-sided interpolant, $\mathbf{x}_t = \alpha_t \mathbf{x}_1 + \gamma_t \mathbf{z}$, $\quad \mathbf{x}_1 \sim p_{\text{data}}$, $\quad \mathbf{z} \sim \mathcal{N}(0, \boldsymbol{I})$ satisfying regularity conditions as stated in [Albergo et al., 2023], and a degraded signal $\mathbf{y}$ generated using Eqn. 2, the conditional velocity field $\boldsymbol{b}(\mathbf{x}_t, \mathbf{y}, t)$ can be approximately estimated from the unconditional velocity field $\boldsymbol{b}_\theta(\mathbf{x}_t, t)$ as [Pokle et al., 2024],

$$\boldsymbol{b}(\mathbf{x}_t, \mathbf{y}, t) \approx \boldsymbol{b}_\theta(\mathbf{x}_t, t) + w_t \frac{\gamma_t}{\alpha_t}\Big[\gamma_t \dot{\alpha}_t - \dot{\gamma}_t \alpha_t\Big]\nabla_{\mathbf{x}_t} \log p(\mathbf{y}|\mathbf{x}_t) \tag{15}$$

where $w_t$ represents a time-dependent scalar guidance schedule and $\dot{\sigma}_t, \dot{\alpha}_t$ represent the first-order time derivatives of $\sigma_t$ and $\alpha_t$, respectively. Our proof consists of two parts: Firstly, we establish the connection between the unconditional velocity field $\boldsymbol{b}(\mathbf{x}_t, t)$ for the one-sided interpolant and the score function $\boldsymbol{s}(\mathbf{x}_t, t)$ associated with the marginal distribution $p(\mathbf{x}_t)$. Secondly, we use this connection to estimate the conditional velocity $\boldsymbol{b}(\mathbf{x}_t, \boldsymbol{y}, t)$ in terms of $\boldsymbol{b}(\mathbf{x}_t, t)$ to establish the required result.

**Connection between $\boldsymbol{b}(\mathbf{x}_t, t)$ and $\boldsymbol{s}(\mathbf{x}_t, t)$:** For the one-sided interpolant, by definition,

$$\mathbf{x}_t = \alpha_t \mathbf{x}_1 + \gamma_t \mathbf{z} \quad \mathbf{x}_1 \sim p_{\text{data}}, \quad \mathbf{z} \sim \mathcal{N}(0, \boldsymbol{I}) \tag{16}$$

Taking the expectation w.r.t $p(\mathbf{x}_1, \mathbf{z})$ on both sides conditioned on the noisy state $\mathbf{x}_t$, we have,

$$\mathbf{x}_t = \alpha_t \mathbb{E}[\mathbf{x}_1|\mathbf{x}_t] + \gamma_t \mathbb{E}[\mathbf{z}|\mathbf{x}_t] \tag{17}$$

Furthermore, we have the following result from Albergo et al. [2023],

$$\mathbb{E}[\mathbf{z}|\mathbf{x}_t] = -\gamma_t \boldsymbol{s}(\mathbf{x}_t, t) \tag{18}$$

where $\boldsymbol{s}(\mathbf{x}_t, t)$ represents the score function. From Eqns. 17, 18, it follows,

$$\mathbb{E}[\mathbf{x}_1|\mathbf{x}_t] = \frac{1}{\alpha_t}\Big[\mathbf{x}_t + \gamma_t^2 \boldsymbol{s}(\mathbf{x}_t, t)\Big] \tag{19}$$

Intuitively, the above result represents Tweedie's estimate [Stein, 1981] for estimating $\hat{\mathbf{x}}_1 = \mathbb{E}[\mathbf{x}_1|\mathbf{x}_t]$ in the context of one-sided stochastic interpolants. Next, the one-sided interpolant also induces an unconditional velocity field specified as:

$$\begin{aligned}\mathbf{b}(\mathbf{x}_t, t) &= \dot{\alpha}_t \mathbb{E}[\mathbf{x}_1|\mathbf{x}_t] + \dot{\gamma}_t \mathbb{E}[\mathbf{z}|\mathbf{x}_t]\\ &= \dot{\alpha}_t \mathbb{E}[\mathbf{x}_1|\mathbf{x}_t] - \dot{\gamma}_t \gamma_t \boldsymbol{s}(\mathbf{x}_t, t)\end{aligned} \tag{20}$$

where $\dot{\gamma}_t, \dot{\alpha}_t$ represent the first-order time derivatives of $\gamma_t$ and $\alpha_t$, respectively. Substituting the result from Eqn. 19 into Eqn. 20, we have the following result,

$$\boldsymbol{b}(\mathbf{x}_t, t) = \frac{\dot{\alpha}_t}{\alpha_t}\mathbf{x}_t + \frac{\gamma_t}{\alpha_t}\Big[\gamma_t \dot{\alpha}_t - \dot{\gamma}_t \alpha_t\Big]\boldsymbol{s}(\mathbf{x}_t, t) \tag{21}$$

This concludes the first part of the proof.

**Estimating the conditional velocity $b(\mathbf{x}_t, y, t)$ in terms of $b(\mathbf{x}_t, t)$.** For transport conditioned on $\mathbf{y}$, the conditional velocity can be expressed as (following the result in Eqn. 21):

$$b(\mathbf{x}_t, y, t) = \frac{\dot{\alpha}_t}{\alpha_t}\mathbf{x}_t + \frac{\gamma_t}{\alpha_t}\Big[\gamma_t\dot{\alpha}_t - \dot{\gamma}_t\alpha_t\Big]s(\mathbf{x}_t, y, t) \tag{22}$$

$$= \frac{\dot{\alpha}_t}{\alpha_t}\mathbf{x}_t + \frac{\gamma_t}{\alpha_t}\Big[\gamma_t\dot{\alpha}_t - \dot{\gamma}_t\alpha_t\Big]\Big[s(\mathbf{x}_t, t) + w_t\nabla_{\mathbf{x}_t}\log p(\mathbf{y}|\mathbf{x}_t)\Big] \tag{23}$$

$$= \frac{\dot{\alpha}_t}{\alpha_t}\mathbf{x}_t + \frac{\gamma_t}{\alpha_t}\Big[\gamma_t\dot{\alpha}_t - \dot{\gamma}_t\alpha_t\Big]s(\mathbf{x}_t, t) + w_t\frac{\gamma_t}{\alpha_t}\Big[\gamma_t\dot{\alpha}_t - \dot{\gamma}_t\alpha_t\Big]\nabla_{\mathbf{x}_t}\log p(\mathbf{y}|\mathbf{x}_t) \tag{24}$$

$$= b(\mathbf{x}_t, t) + w_t\frac{\gamma_t}{\alpha_t}\Big[\gamma_t\dot{\alpha}_t - \dot{\gamma}_t\alpha_t\Big]\nabla_{\mathbf{x}_t}\log p(\mathbf{y}|\mathbf{x}_t) \tag{25}$$

Approximating the unconditional velocity $b(\mathbf{x}_t, t)$ using a parametric estimator $b_\theta(\mathbf{x}_t, t)$, we get the required result.

$$b(\mathbf{x}_t, y, t) \approx b_\theta(\mathbf{x}_t, t) + w_t\frac{\gamma_t}{\alpha_t}\Big[\gamma_t\dot{\alpha}_t - \dot{\gamma}_t\alpha_t\Big]\nabla_{\mathbf{x}_t}\log p(\mathbf{y}|\mathbf{x}_t) \tag{26}$$

$$\square$$

## A.2 Proof of Proposition 1

We restate Proposition 1 for convenience,

**Proposition.** *For the conditional diffusion dynamics defined in Eqn. 3, introducing the transformation $\bar{\mathbf{x}}_t = A_t\mathbf{x}_t$ induces the following projected diffusion dynamics.*

$$d\hat{\mathbf{x}}_t = A_tB_tA_t^{-1}\hat{\mathbf{x}}_t dt + d\Phi_t\epsilon_\theta(\mathbf{x}_t, t) - \frac{w_t r_t^{-2}}{2}G_tG_t^\top\frac{\partial\hat{\mathbf{x}}_1}{\partial\mathbf{x}_t}^\top(H^\dagger y - P\hat{\mathbf{x}}_1)dt \tag{27}$$

$$A_t = \exp\Big(\int_0^t B_s - F_s ds\Big), \qquad \Phi_t = -\int_0^t \frac{1}{2}A_sG_sG_s^\top C_{out}(s)ds, \tag{28}$$

*where $H^\dagger = H^\top(HH^\top)^{-1}$ and $P = H^\top(HH^\top)^{-1}H$ represent the pseudoinverse and the orthogonal projector operators for the degradation operator $H$.*

*Proof.* We have the following form of the conditional diffusion dynamics

$$\frac{d\mathbf{x}_t}{dt} = F_t\mathbf{x}_t - \frac{1}{2}G_tG_t^\top\nabla_{\mathbf{x}_t}\log p(\mathbf{x}_t|y) \tag{29}$$

$$= F_t\mathbf{x}_t - \frac{1}{2}G_tG_t^\top s_\theta(\mathbf{x}_t, t) - \frac{1}{2}w_tG_tG_t^\top\nabla_{\mathbf{x}_t}\log p(\mathbf{y}|\mathbf{x}_t) \tag{30}$$

Given an affine transformation which projects the state $\mathbf{x}_t$ to $\hat{\mathbf{x}}_t$,

$$\hat{\mathbf{x}}_t = A_t\mathbf{x}_t \tag{31}$$

Therefore, by the Chain Rule of calculus,

$$\frac{d\hat{\mathbf{x}}_t}{dt} = \frac{dA_t}{dt}\mathbf{x}_t + A_t\frac{d\mathbf{x}_t}{dt} \tag{32}$$

Substituting the ODE in Eqn. 30 in Eqn. 32

$$\frac{d\hat{\mathbf{x}}_t}{dt} = \frac{dA_t}{dt}\mathbf{x}_t + A_t\Big[F_t\mathbf{x}_t - \frac{1}{2}G_tG_t^\top\nabla_{\mathbf{x}_t}s_\theta(\mathbf{x}_t, t) - \frac{1}{2}w_tG_tG_t^\top\nabla_{\mathbf{x}_t}\log p(\mathbf{y}|\mathbf{x}_t)\Big] \tag{33}$$

$$= \Big[\frac{dA_t}{dt} + A_tF_t\Big]\mathbf{x}_t - \frac{1}{2}A_tG_tG_t^\top s_\theta(\mathbf{x}_t, t) - \frac{1}{2}w_tA_tG_tG_t^\top\nabla_{\mathbf{x}_t}\log p(\mathbf{y}|\mathbf{x}_t) \tag{34}$$

Since, we have $s_\theta(\mathbf{x}_t, t) = C_{out}(t)\epsilon_\theta(\mathbf{x}_t, t)$, the above equation can be simplified as,

$$\frac{d\hat{\mathbf{x}}_t}{dt} = \Big[\frac{dA_t}{dt} + A_tF_t\Big]\mathbf{x}_t - \frac{1}{2}A_tG_tG_t^\top C_{out}(t)\epsilon_\theta(\mathbf{x}_t, t) - \frac{1}{2}w_tA_tG_tG_t^\top\nabla_{\mathbf{x}_t}\log p(\mathbf{y}|\mathbf{x}_t)\Big] \tag{35}$$

Further parameterizing,

$$\frac{d\boldsymbol{A}_t}{dt} + \boldsymbol{A}_t \boldsymbol{F}_t = \boldsymbol{A}_t \boldsymbol{B}_t \tag{36}$$

$$\frac{d\boldsymbol{\Phi}_t}{dt} = -\frac{1}{2} \boldsymbol{A}_t \boldsymbol{G}_t \boldsymbol{G}_t^\top \boldsymbol{C}_{\text{out}}(t) \tag{37}$$

which yields the required diffusion ODE in the projected space:

$$d\hat{\mathbf{x}}_t = \boldsymbol{A}_t \boldsymbol{B}_t \boldsymbol{A}_t^{-1} \hat{\mathbf{x}}_t dt + d\boldsymbol{\Phi}_t \boldsymbol{\epsilon}_{\boldsymbol{\theta}}(\mathbf{x}_t, t) - \frac{1}{2} w_t \boldsymbol{A}_t \boldsymbol{G}_t \boldsymbol{G}_t^\top \nabla_{\mathbf{x}_t} \log p(\mathbf{y}|\mathbf{x}_t) dt \tag{38}$$

We have the following approximation for the conditional score $\nabla_{\mathbf{x}_t} \log p(\mathbf{y}|\mathbf{x}_t)$ (for the noiseless case $\sigma_y = 0$)

$$\nabla_{\mathbf{x}_t} \log p(\mathbf{y}|\mathbf{x}_t) = r_t^{-2} \frac{\partial \hat{\mathbf{x}}_0}{\partial \mathbf{x}_t}^\top \boldsymbol{H}^\top (\boldsymbol{H}\boldsymbol{H}^\top)^{-1}(\boldsymbol{y} - \boldsymbol{H}\hat{\mathbf{x}}_0) \tag{39}$$

$$= r_t^{-2} \frac{\partial \hat{\mathbf{x}}_0}{\partial \mathbf{x}_t}^\top (\boldsymbol{H}^\top (\boldsymbol{H}\boldsymbol{H}^\top)^{-1}\boldsymbol{y} - \boldsymbol{H}^\top (\boldsymbol{H}\boldsymbol{H}^\top)^{-1}\boldsymbol{H}\hat{\mathbf{x}}_0) \tag{40}$$

$$= r_t^{-2} \frac{\partial \hat{\mathbf{x}}_0}{\partial \mathbf{x}_t}^\top (\boldsymbol{H}^\dagger \boldsymbol{y} - \boldsymbol{P}\hat{\mathbf{x}}_0) \tag{41}$$

where $\boldsymbol{H}^\dagger = \boldsymbol{H}^\top (\boldsymbol{H}\boldsymbol{H}^\top)^{-1}$ and $\boldsymbol{P} = \boldsymbol{H}^\top (\boldsymbol{H}\boldsymbol{H}^\top)^{-1}\boldsymbol{H}$ represent the pseudoinverse and the orthogonal projector operators for the degradation operator $\boldsymbol{H}$. Substituting this form of the conditional score in projected diffusion dynamics, we have,

$$d\hat{\mathbf{x}}_t = \boldsymbol{A}_t \boldsymbol{B}_t \boldsymbol{A}_t^{-1} \hat{\mathbf{x}}_t dt + d\boldsymbol{\Phi}_t \boldsymbol{\epsilon}_{\boldsymbol{\theta}}(\mathbf{x}_t, t) - \frac{w_t r_t^{-2}}{2} \boldsymbol{A}_t \boldsymbol{G}_t \boldsymbol{G}_t^\top \frac{\partial \hat{\mathbf{x}}_0}{\partial \mathbf{x}_t}^\top (\boldsymbol{H}^\dagger \boldsymbol{y} - \boldsymbol{P}\hat{\mathbf{x}}_0) dt \tag{42}$$

This concludes the proof. $\qquad \square$

### A.3  Proof of Proposition 2

We restate the theorem here for convenience.

**Proposition.** *For a noiseless linear inverse problem with $\sigma_y = 0$ and the conditional score approximated using Eqn. 5, introducing the transformation $\bar{\mathbf{x}}_t = \boldsymbol{A}_t \mathbf{x}_t$ also induces the following projected diffusion dynamics.*

$$d\bar{\mathbf{x}}_t = \boldsymbol{A}_t \boldsymbol{B}_t \boldsymbol{A}_t^{-1} \bar{\mathbf{x}}_t + d\boldsymbol{\Phi}_y \boldsymbol{y} + d\boldsymbol{\Phi}_s \boldsymbol{\epsilon}_\theta(\mathbf{x}_t, t) + d\boldsymbol{\Phi}_j \left[ \partial_{\mathbf{x}_t} \boldsymbol{\epsilon}_\theta(\mathbf{x}_t, t)(\boldsymbol{H}^\dagger \boldsymbol{y} - \boldsymbol{P}\hat{\mathbf{x}}_1) \right] \tag{43}$$

$$\boldsymbol{A}_t = \exp\left[ \int_0^t \boldsymbol{B}_s - \left( \boldsymbol{F}_s + \frac{w_s r_s^{-2}}{2\mu_s^2} \boldsymbol{G}_s \boldsymbol{G}_s^\top \boldsymbol{P} \right) ds \right] \qquad d\boldsymbol{\Phi}_y = -\frac{w_t r_t^{-2}}{2\mu_t} \boldsymbol{A}_t \boldsymbol{G}_t \boldsymbol{G}_t^\top \boldsymbol{H}^\dagger \tag{44}$$

$$d\boldsymbol{\Phi}_s = -\frac{1}{2} \boldsymbol{A}_t \boldsymbol{G}_t \boldsymbol{G}_t^\top \left[ \boldsymbol{I}_d - \frac{w_t r_t^{-2} \sigma_t^2}{\mu_t^2} \boldsymbol{A}_t \boldsymbol{P} \right] \boldsymbol{C}_{\text{out}}(t) \qquad d\boldsymbol{\Phi}_j = -\frac{w_t r_t^{-2} \sigma_t^2}{2\mu_t} \boldsymbol{A}_t \boldsymbol{G}_t \boldsymbol{G}_t^\top \boldsymbol{C}_{\text{out}}(t) \tag{45}$$

*where $\exp(.)$ denotes the matrix exponential, $\boldsymbol{H}^\dagger$, and $\boldsymbol{P}$ are the pseudoinverse and projector operators (as defined previously).*

*Proof.* The proof consists of two parts. Firstly, we simplify the conditional score $\nabla_{\mathbf{x}_t} \log p(\mathbf{y}|\mathbf{x}_t)$. Secondly, we plug the simplified form of the conditional score into the conditional diffusion dynamics and develop conjugate integrators.

**Exploiting the linearity in $\nabla_{\mathbf{x}_t} \log p(\mathbf{y}|\mathbf{x}_t)$:** From the definition of the score $\nabla_{\mathbf{x}_t} \log p(\mathbf{y}|\mathbf{x}_t)$:

$$\nabla_{\mathbf{x}_t} \log p(\mathbf{y}|\mathbf{x}_t) = \frac{\partial \hat{\mathbf{x}}_0}{\partial \mathbf{x}_t}^\top \boldsymbol{H}^\top \boldsymbol{\Sigma}_t^{-1} (\boldsymbol{y} - \boldsymbol{H}\hat{\mathbf{x}}_0) \tag{46}$$

where $\hat{\mathbf{x}}_0$ is the Tweedie's estimate of $\mathbb{E}(\mathbf{x}_1|\mathbf{x}_t)$ given by:

$$\hat{\mathbf{x}}_0 = \frac{1}{\mu_t}(\mathbf{x}_t + \sigma_t^2 \boldsymbol{s}_\theta(\mathbf{x}_t, t)) \tag{47}$$

where $\mu_t$, $\sigma_t$ are the mean coefficient and standard deviation of the perturbation kernel $p(\mathbf{x}_t|\mathbf{x}_1) = \mathcal{N}(\mu_t\mathbf{x}_1\sigma_t^2\boldsymbol{I}_d)$, respectively, and $\boldsymbol{\Sigma}_t = r_t^2(\boldsymbol{H}\boldsymbol{H}^\top)$ is the variance of the ΠGDM approximation of $p(\mathbf{y}|\mathbf{x}_t)$ (for the noiseless case i.e. $\sigma_y = 0$). Therefore,

$$\nabla_{\mathbf{x}_t} \log p(\mathbf{y}|\mathbf{x}_t) = \frac{\partial \hat{\mathbf{x}}_0}{\partial \mathbf{x}_t}^\top \boldsymbol{H}^\top \boldsymbol{\Sigma}_t^{-1}(\boldsymbol{y} - \boldsymbol{H}\hat{\mathbf{x}}_0) \tag{48}$$

$$= \frac{1}{\mu_t}(\boldsymbol{I}_d + \sigma_t^2 \underbrace{\nabla_{\mathbf{x}_t}\boldsymbol{s}_\theta(\mathbf{x}_t,t)}_{=\boldsymbol{S}_\theta(\mathbf{x}_t,t)})\boldsymbol{H}^\top \boldsymbol{\Sigma}_t^{-1}(\boldsymbol{y} - \boldsymbol{H}\hat{\mathbf{x}}_0) \tag{49}$$

$$= \frac{1}{\mu_t}\left[\boldsymbol{H}^\top \boldsymbol{\Sigma}_t^{-1}(\boldsymbol{y} - \boldsymbol{H}\hat{\mathbf{x}}_0) + \sigma_t^2 \boldsymbol{S}_\theta(\mathbf{x}_t,t)\boldsymbol{H}^\top \boldsymbol{\Sigma}_t^{-1}(\boldsymbol{y} - \boldsymbol{H}\hat{\mathbf{x}}_0)\right] \tag{50}$$

$$= \frac{1}{\mu_t}\left[\boldsymbol{H}^\top \boldsymbol{\Sigma}_t^{-1}(\boldsymbol{y} - \frac{1}{\mu_t}\boldsymbol{H}(\mathbf{x}_t + \sigma_t^2 \boldsymbol{s}_\theta(\mathbf{x}_t,t))) + \sigma_t^2 \boldsymbol{S}_\theta(\mathbf{x}_t,t)\boldsymbol{H}^\top \boldsymbol{\Sigma}_t^{-1}(\boldsymbol{y} - \boldsymbol{H}\hat{\mathbf{x}}_0)\right] \tag{51}$$

$$= \underbrace{\frac{1}{\mu_t}\boldsymbol{H}^\top \boldsymbol{\Sigma}_t^{-1}\boldsymbol{y} - \frac{1}{\mu_t^2}\boldsymbol{H}^\top \boldsymbol{\Sigma}_t^{-1}\boldsymbol{H}\mathbf{x}_t}_{\text{Linear Terms}} \tag{52}$$

$$\underbrace{- \frac{\sigma_t^2}{\mu_t^2}\boldsymbol{H}^\top \boldsymbol{\Sigma}_t^{-1}\boldsymbol{H}\boldsymbol{s}_\theta(\mathbf{x}_t,t)) + \frac{\sigma_t^2}{\mu_t}\boldsymbol{S}_\theta(\mathbf{x}_t,t)\boldsymbol{H}^\top \boldsymbol{\Sigma}_t^{-1}(\boldsymbol{y} - \boldsymbol{H}\hat{\mathbf{x}}_0)}_{\text{Non-Linear Terms}} \tag{53}$$

where $\boldsymbol{S}_\theta$ denotes the second-order derivative of the score function $\boldsymbol{s}_\theta(\mathbf{x}_t,t)$. Therefore, the conditional score $\nabla_{\mathbf{x}_t} \log p(\mathbf{y}|\mathbf{x}_t)$, can be decomposed into a combination of linear and non-linear terms. Next, we use this decomposition to design conjugate integrators for noiseless linear inverse problems.

**Conjugate Integrator Design:** From Eqn. 30, the conditional reverse diffusion dynamics can be specified as:

$$\frac{d\mathbf{x}_t}{dt} = \boldsymbol{F}_t\mathbf{x}_t - \frac{1}{2}\boldsymbol{G}_t\boldsymbol{G}_t^\top \boldsymbol{s}_\theta(\mathbf{x}_t,t) - \frac{1}{2}w_t\boldsymbol{G}_t\boldsymbol{G}_t^\top \nabla_{\mathbf{x}_t} \log p(\mathbf{y}|\mathbf{x}_t) \tag{54}$$

Plugging in the form of the conditional score in Eqn. 53 in the above equation, we have,

$$\frac{d\mathbf{x}_t}{dt} = \boldsymbol{F}_t\mathbf{x}_t - \frac{1}{2}\boldsymbol{G}_t\boldsymbol{G}_t^\top \boldsymbol{s}_\theta(\mathbf{x}_t,t) - \frac{1}{2}w_t\boldsymbol{G}_t\boldsymbol{G}_t^\top \nabla_{\mathbf{x}_t} \log p(\mathbf{y}|\mathbf{x}_t) \tag{55}$$

$$= \boldsymbol{F}_t\mathbf{x}_t - \frac{1}{2}\boldsymbol{G}_t\boldsymbol{G}_t^\top \boldsymbol{s}_\theta(\mathbf{x}_t,t) - \frac{1}{2}w_t\boldsymbol{G}_t\boldsymbol{G}_t^\top \left[\frac{1}{\mu_t}\boldsymbol{H}^\top \boldsymbol{\Sigma}_t^{-1}\boldsymbol{y} - \frac{1}{\mu_t^2}\boldsymbol{H}^\top \boldsymbol{\Sigma}_t^{-1}\boldsymbol{H}\mathbf{x}_t \right. \tag{56}$$

$$\left. - \frac{\sigma_t^2}{\mu_t^2}\boldsymbol{H}^\top \boldsymbol{\Sigma}_t^{-1}\boldsymbol{H}\boldsymbol{s}_\theta(\mathbf{x}_t,t)) + \frac{\sigma_t^2}{\mu_t}\boldsymbol{S}_\theta(\mathbf{x}_t,t)\boldsymbol{H}^\top \boldsymbol{\Sigma}_t^{-1}(\boldsymbol{y} - \boldsymbol{H}\hat{\mathbf{x}}_0)\right] \tag{57}$$

$$= \left[\boldsymbol{F}_t + \frac{w_t}{2\mu_t^2}\boldsymbol{G}_t\boldsymbol{G}_t^\top \boldsymbol{H}^\top \boldsymbol{\Sigma}_t^{-1}\boldsymbol{H}\right]\mathbf{x}_t - \frac{w_t}{2\mu_t}\boldsymbol{G}_t\boldsymbol{G}_t^\top \boldsymbol{H}^\top \boldsymbol{\Sigma}_t^{-1}\boldsymbol{y} \tag{58}$$

$$- \frac{1}{2}\boldsymbol{G}_t\boldsymbol{G}_t^\top \left[\boldsymbol{I}_d - \frac{w_t\sigma_t^2}{\mu_t^2}\boldsymbol{H}^\top \boldsymbol{\Sigma}_t^{-1}\boldsymbol{H}\right]\boldsymbol{s}_\theta(\mathbf{x}_t,t) - \frac{w_t\sigma_t^2}{2\mu_t}\boldsymbol{G}_t\boldsymbol{G}_t^\top \left[\boldsymbol{S}_\theta(\mathbf{x}_t,t)\boldsymbol{H}^\top \boldsymbol{\Sigma}_t^{-1}(\boldsymbol{y} - \boldsymbol{H}\hat{\mathbf{x}}_0)\right] \tag{59}$$

Given an affine transformation which projects the state $\mathbf{x}_t$ to $\hat{\mathbf{x}}_t$,

$$\hat{\mathbf{x}}_t = \boldsymbol{A}_t\mathbf{x}_t \tag{60}$$

the projected diffusion dynamics can be specified as:

$$\frac{d\bar{\mathbf{x}}_t}{dt} = \left[\frac{d\boldsymbol{A}_t}{dt} + \boldsymbol{A}_t(\boldsymbol{F}_t + \frac{w_t}{2\mu_t^2}\boldsymbol{G}_t\boldsymbol{G}_t^\top \boldsymbol{H}^\top \boldsymbol{\Sigma}_t^{-1}\boldsymbol{H})\right]\mathbf{x}_t - \frac{w_t}{2\mu_t}\boldsymbol{A}_t\boldsymbol{G}_t\boldsymbol{G}_t^\top \boldsymbol{H}^\top \boldsymbol{\Sigma}_t^{-1}\boldsymbol{y} \tag{61}$$

$$- \frac{1}{2}\boldsymbol{A}_t\boldsymbol{G}_t\boldsymbol{G}_t^\top \left[\boldsymbol{I}_d - \frac{w_t\sigma_t^2}{\mu_t^2}\boldsymbol{A}_t\boldsymbol{H}^\top \boldsymbol{\Sigma}_t^{-1}\boldsymbol{H}\right]\boldsymbol{s}_\theta(\mathbf{x}_t,t) - \frac{w_t\sigma_t^2}{2\mu_t}\boldsymbol{A}_t\boldsymbol{G}_t\boldsymbol{G}_t^\top \left[\boldsymbol{S}_\theta(\mathbf{x}_t,t)\boldsymbol{H}^\top \boldsymbol{\Sigma}_t^{-1}(\boldsymbol{y} - \boldsymbol{H}\hat{\mathbf{x}}_1)\right] \tag{62}$$

Furthermore, the score network is parameterized as $\boldsymbol{s}_\theta(\mathbf{x}_t,t) = \boldsymbol{C}_{\text{out}}(t)\boldsymbol{\epsilon}_\theta(\mathbf{x}_t,t)$. Consequently, $\boldsymbol{S}_\theta(\mathbf{x}_t,t) = \boldsymbol{C}_{\text{out}}(t)\partial_t\boldsymbol{\epsilon}_\theta(\mathbf{x}_t,t)$. Lastly, we reparameterize $\boldsymbol{\Sigma}_t = r_t^2\boldsymbol{\Sigma}$ where $\boldsymbol{\Sigma} = \boldsymbol{H}\boldsymbol{H}^\top$. Plugging

these parameterizations in the projected diffusion dynamics, we have,

$$\frac{d\bar{\mathbf{x}}_t}{dt} = \Big[\frac{d\boldsymbol{A}_t}{dt} + \boldsymbol{A}_t(\boldsymbol{F}_t + \frac{w_t r_t^{-2}}{2\mu_t^2}\boldsymbol{G}_t\boldsymbol{G}_t^\top \boldsymbol{H}^\top \boldsymbol{\Sigma}^{-1}\boldsymbol{H})\Big]\mathbf{x}_t - \frac{w_t r_t^{-2}}{2\mu_t}\boldsymbol{A}_t\boldsymbol{G}_t\boldsymbol{G}_t^\top \boldsymbol{H}^\top \boldsymbol{\Sigma}^{-1}\boldsymbol{y} \quad (63)$$

$$- \frac{1}{2}\boldsymbol{A}_t\boldsymbol{G}_t\boldsymbol{G}_t^\top \Big[\boldsymbol{I}_d - \frac{w_t r_t^{-2}\sigma_t^2}{\mu_t^2}\boldsymbol{A}_t\boldsymbol{H}^\top \boldsymbol{\Sigma}^{-1}\boldsymbol{H}\Big]\boldsymbol{C}_{\text{out}}(t)\boldsymbol{\epsilon}_\theta(\mathbf{x}_t, t) \quad (64)$$

$$- \frac{w_t r_t^{-2}\sigma_t^2}{2\mu_t}\boldsymbol{A}_t\boldsymbol{G}_t\boldsymbol{G}_t^\top \boldsymbol{C}_{\text{out}}(t)\Big[\partial_{\mathbf{x}_t}\boldsymbol{\epsilon}_\theta(\mathbf{x}_t, t)\boldsymbol{H}^\top \boldsymbol{\Sigma}^{-1}(\boldsymbol{y} - \boldsymbol{H}\hat{\mathbf{x}}_0)\Big] \quad (65)$$

We then parameterize,

$$\frac{d\boldsymbol{A}_t}{dt} + \boldsymbol{A}_t\Big(\boldsymbol{F}_t + \frac{w_t r_t^{-2}}{2\mu_t^2}\boldsymbol{G}_t\boldsymbol{G}_t^\top \boldsymbol{H}^\top \boldsymbol{\Sigma}_t^{-1}\boldsymbol{H}\Big) = \boldsymbol{A}_t\boldsymbol{B}_t \quad (66)$$

This implies,

$$\boldsymbol{A}_t = \exp\Big[\int_0^t \boldsymbol{B}_s - \Big(\boldsymbol{F}_s + \frac{w_s r_s^{-2}}{2\mu_s^2}\boldsymbol{G}_s\boldsymbol{G}_s^\top \boldsymbol{P}\Big)ds\Big] \quad (67)$$

where $\exp(.)$ denotes the matrix exponential. Furthermore, we parameterize,

$$\boldsymbol{\Phi}_y = -\int_0^t \frac{w_s r_s^{-2}}{2\mu_s}\boldsymbol{A}_s\boldsymbol{G}_s\boldsymbol{G}_s^\top \boldsymbol{H}^\dagger ds \quad (68)$$

$$\boldsymbol{\Phi}_s = -\int_0^t \frac{1}{2}\boldsymbol{A}_s\boldsymbol{G}_s\boldsymbol{G}_s^\top \Big[\boldsymbol{I}_d - \frac{w_s r_s^{-2}\sigma_s^2}{\mu_s^2}\boldsymbol{A}_s\boldsymbol{P}\Big]\boldsymbol{C}_{\text{out}}(s)ds \quad (69)$$

$$\boldsymbol{\Phi}_j = -\int_0^t \frac{w_s r_s^{-2}\sigma_s^2}{2\mu_s}\boldsymbol{A}_s\boldsymbol{G}_s\boldsymbol{G}_s^\top \boldsymbol{C}_{\text{out}}(s)ds \quad (70)$$

With this parameterization, the projected diffusion dynamics can be compactly specified as follows:

$$d\bar{\mathbf{x}}_t = \boldsymbol{A}_t\boldsymbol{B}_t\boldsymbol{A}_t^{-1}\bar{\mathbf{x}}_t + d\boldsymbol{\Phi}_y\boldsymbol{y} + d\boldsymbol{\Phi}_s\boldsymbol{\epsilon}_\theta(\mathbf{x}_t, t) + d\boldsymbol{\Phi}_j\Big[\partial_{\mathbf{x}_t}\boldsymbol{\epsilon}_\theta(\mathbf{x}_t, t)(\boldsymbol{H}^\dagger\boldsymbol{y} - \boldsymbol{P}\hat{\mathbf{x}}_0)\Big] \quad (71)$$

This concludes the proof. $\qquad\square$

### A.4 Simplification in Eqn. 11

We restate the result for convenience. The matrix exponential in Eqn. 10

$$\boldsymbol{A}_t = \exp\Big[\int_0^t \Big(\lambda + \frac{1}{2}\beta_s\Big)ds\,\boldsymbol{I}_d - \frac{w}{2}\Big(\int_0^t \beta_s ds\Big)\boldsymbol{P}\Big] \quad (72)$$

can be simplified as,

$$\boldsymbol{A}_t = \exp(\kappa_t^1)\Big[\boldsymbol{I}_d + (\exp(\kappa_t^2) - 1)\boldsymbol{P}\Big], \quad \kappa_t^1 = \int_0^t \Big(\lambda + \frac{1}{2}\beta_s\Big)ds, \quad \kappa_t^2 = -\frac{w}{2}\int_0^t \beta_s ds \quad (73)$$

where $\boldsymbol{P}$ is the orthogonal projector corresponding to the degradation operator $\boldsymbol{H}$.

*Proof.* We have,

$$\boldsymbol{A}_t = \exp(\kappa_t^1\boldsymbol{I}_d + \kappa_t^2\boldsymbol{P}) = \exp(\kappa_t^1\,\boldsymbol{I}_d)\exp(\kappa_t^2\boldsymbol{P}) \quad (74)$$

The above result follows since $\boldsymbol{I}_d\boldsymbol{P} = \boldsymbol{P}\boldsymbol{I}_d$ (commutative under multiplication). Moreover, we can further simplify the matrix exponential in $\boldsymbol{P}$ as follows,

$$\exp(\kappa_t^2\boldsymbol{P}) = \sum_{i=0}^\infty \frac{(\kappa_t^2)^i\boldsymbol{P}^i}{i!} \quad (75)$$

$$= \boldsymbol{I}_d + \sum_{i=1}^\infty \frac{(\kappa_t^2)^i\boldsymbol{P}^i}{i!} = \boldsymbol{I}_d + \Big[\sum_{i=1}^\infty \frac{(\kappa_t^2)^i}{i!}\Big]\boldsymbol{P} \quad (76)$$

The above result follows from the property of orthogonal projectors $P^2 = P$. Therefore,

$$\exp(\kappa_t^2\boldsymbol{P}) = \boldsymbol{I}_d + \Big[\sum_{i=1}^\infty \frac{(\kappa_t^2)^i}{i!}\Big]\boldsymbol{P} = \boldsymbol{I}_d + \Big[\sum_{i=0}^\infty \frac{(\kappa_t^2)^i}{i!} - 1\Big]\boldsymbol{P} \quad (77)$$

$$\exp(\kappa_t^2\boldsymbol{P}) = \boldsymbol{I}_d + \Big[\exp(\kappa_t^2) - 1\Big]\boldsymbol{P} \quad (78)$$

which concludes the proof. $\qquad\square$

### A.5 Proof of Proposition for Solving Noisy Inverse Problems

We restate the result here for convenience.

**Proposition.** *For the noisy inverse problem,*

$$\boldsymbol{y} = \boldsymbol{H}\mathbf{x}_0 + \sigma_y \mathbf{z}, \quad \mathbf{z} \sim \mathcal{N}(0, \boldsymbol{I}_d), \tag{79}$$

*given the transformation $\boldsymbol{A}_t$ for the noiseless case as defined in Eqn. 11, the corresponding noisy transformation can be approximated as,*

$$\boldsymbol{A}_t^{\sigma_y} = \boldsymbol{A}_t + \kappa_3(t) \boldsymbol{H}^\dagger (\boldsymbol{H}^\dagger)^\top + \mathcal{O}(\sigma_y^4) \tag{80}$$

$$\kappa_3(t) = \frac{w\sigma_y^2}{2} \Big( \int_0^t \frac{\beta_s}{r_s^2} ds \Big) \Big[ \exp\Big( \kappa_1(t) + \kappa_2(t) \Big) - 1 \Big] \tag{81}$$

*Consequently, the inverse of the transformation $\boldsymbol{A}_t^{\sigma_y}$ can be approximated as,*

$$(\boldsymbol{A}_t^{\sigma_y})^{-1} \approx \boldsymbol{A}_t^{-1} - \kappa_3(t) \boldsymbol{A}_t^{-1} \boldsymbol{H}^\dagger (\boldsymbol{H}^\dagger)^\top \boldsymbol{A}_t^{-1} + \mathcal{O}(\sigma_y^4) \tag{82}$$

*Proof.* We have,

$$\boldsymbol{A}_t^{\sigma_y} = \exp\Big[ \int_0^t \Big( \lambda + \frac{1}{2}\beta_s \Big) ds - \frac{w}{2} \Big( \int_0^t \beta_s \boldsymbol{H}^\top (\boldsymbol{H}\boldsymbol{H}^\top + \frac{\sigma_y^2}{r_t^2} \boldsymbol{I}_d)^{-1} \boldsymbol{H} ds \Big) \Big] \tag{83}$$

From perturbation analysis, we introduce the following first-order approximation,

$$\boldsymbol{H}^\top (\boldsymbol{H}\boldsymbol{H}^\top + \frac{\sigma_y^2}{r_t^2} \boldsymbol{I}_d)^{-1} \boldsymbol{H} \approx \boldsymbol{H}^\top \Big[ (\boldsymbol{H}\boldsymbol{H}^\top)^{-1} - \frac{\sigma_y^2}{r_t^2} (\boldsymbol{H}\boldsymbol{H}^\top)^{-2} \Big] \boldsymbol{H} + \mathcal{O}(\sigma_y^4) \tag{84}$$

$$\approx \boldsymbol{H}^\top (\boldsymbol{H}\boldsymbol{H}^\top)^{-1} \boldsymbol{H} - \frac{\sigma_y^2}{r_t^2} \boldsymbol{H}^\top (\boldsymbol{H}\boldsymbol{H}^\top)^{-2} \boldsymbol{H} + \mathcal{O}(\sigma_y^4) \tag{85}$$

$$\approx \boldsymbol{P} - \frac{\sigma_y^2}{r_t^2} \boldsymbol{H}^\dagger (\boldsymbol{H}^\dagger)^\top \tag{86}$$

Substituting this approximation in the expression for $\boldsymbol{A}_t^{\sigma_y}$ (and ignoring terms in $\mathcal{O}(\sigma_y^4)$),

$$\boldsymbol{A}_t^{\sigma_y} \approx \exp\Big[ \int_0^t \Big( \lambda + \frac{1}{2}\beta_s \Big) ds \boldsymbol{I}_d - \frac{w}{2} \Big( \int_0^t \beta_s \Big[ \boldsymbol{P} - \frac{\sigma_y^2}{r_s^2} \boldsymbol{H}^\dagger (\boldsymbol{H}^\dagger)^\top \Big] ds \Big) \Big] \tag{87}$$

$$= \exp\Big[ \underbrace{\int_0^t \Big( \lambda + \frac{1}{2}\beta_s \Big) ds}_{=\kappa_1(t)} \boldsymbol{I}_d \underbrace{- \frac{w}{2} \Big( \int_0^t \beta_s ds \Big)}_{=\kappa_2(t)} \boldsymbol{P} + \frac{w\sigma_y^2}{2} \Big( \int_0^t \frac{\beta_s}{r_s^2} ds \Big) \boldsymbol{H}^\dagger (\boldsymbol{H}^\dagger)^\top \Big] \tag{88}$$

$$= \exp\Big[ \kappa_1(t)\boldsymbol{I}_d + \kappa_2(t)\boldsymbol{P} + \frac{w\sigma_y^2}{2} \Big( \int_0^t \frac{\beta_s}{r_s^2} ds \Big) \boldsymbol{H}^\dagger (\boldsymbol{H}^\dagger)^\top \Big] \tag{89}$$

From the definition of the matrix exponential, it can be shown that the $\boldsymbol{A}_t^{\sigma_y}$ in Eqn. 89 can be approximated as:

$$\boldsymbol{A}_t^{\sigma_y} \approx \exp\Big[ \kappa_1(t)\boldsymbol{I}_d + \kappa_2(t)\boldsymbol{P} \Big] + \underbrace{\frac{w\sigma_y^2}{2} \Big( \int_0^t \frac{\beta_s}{r_s^2} ds \Big) \Big[ \exp\Big( \kappa_1(t) + \kappa_2(t) \Big) - 1 \Big]}_{=\kappa_3(t)} \boldsymbol{H}^\dagger (\boldsymbol{H}^\dagger)^\top + \mathcal{O}(\sigma_y^4)$$

$$\tag{90}$$

Ignoring the higher-order terms, we have,

$$\boldsymbol{A}_t^{\sigma_y} \approx \boldsymbol{A}_t + \kappa_3(t) \boldsymbol{H}^\dagger (\boldsymbol{H}^\dagger)^\top \tag{91}$$

Consequently, we can also approximate the inverse of $\boldsymbol{A}_t^{\sigma_y}$, as follows,

$$(\boldsymbol{A}_t^{\sigma_y})^{-1} = [\boldsymbol{A}_t + \kappa_3(t) \boldsymbol{H}^\dagger (\boldsymbol{H}^\dagger)^\top]^{-1} \tag{92}$$

$$\approx \boldsymbol{A}_t^{-1} - \kappa_3(t) \boldsymbol{A}_t^{-1} \boldsymbol{H}^\dagger (\boldsymbol{H}^\dagger)^\top \boldsymbol{A}_t^{-1} + \mathcal{O}(\sigma_y^4) \tag{93}$$

which concludes the proof. $\square$

# B Conditional Conjugate Integrators: Flows

## B.1 Background

This section discusses conditional conjugate integrators in the context of flows. For brevity, we skip deriving our results for flows since the derivations can be similar to the analysis of diffusion models with minor parameterization changes. Recall that the conditional dynamics for flows are specified as follows [Pokle et al., 2024]:

$$\boldsymbol{b}(\mathbf{x}_t, \mathbf{y}, t) \approx \boldsymbol{b}_\theta(\mathbf{x}_t, t) + w_t \frac{\gamma_t}{\alpha_t} \Big[ \gamma_t \dot{\alpha}_t - \dot{\gamma}_t \alpha_t \Big] \nabla_{\mathbf{x}_t} \log p(\mathbf{y}|\mathbf{x}_t) \tag{94}$$

where $\boldsymbol{b}_\theta(\mathbf{x}_t, t)$ represents the pre-trained velocity field for a flow. Moreover, we restate the form of the conditional score $\nabla_{\mathbf{x}_t} \log p(\mathbf{y}|\mathbf{x}_t)$ for convenience.

$$\nabla_{\mathbf{x}_t} \log p(\mathbf{y}|\mathbf{x}_t) = \frac{\partial \hat{\mathbf{x}}_1}{\partial \mathbf{x}_t}^\top \boldsymbol{H}^\top (r_t^2 \boldsymbol{H}\boldsymbol{H}^\top + \sigma_y^2 \boldsymbol{I}_d)^{-1}(\boldsymbol{y} - \boldsymbol{H}\hat{\mathbf{x}}_1) \tag{95}$$

where $\hat{\mathbf{x}}_1$ represents the Tweedie's estimate of the first moment of $\mathbb{E}(\mathbf{x}_t|\mathbf{x}_1)$,

$$\hat{\mathbf{x}}_1 = \mathbb{E}[\mathbf{x}_1|\mathbf{x}_t] = \frac{1}{\alpha_t} \Big[ \mathbf{x}_t + \gamma_t^2 \boldsymbol{s}(\mathbf{x}_t, t) \Big] \tag{96}$$

where $\boldsymbol{s}(\mathbf{x}_t, t)$ represents the score function associated with the marginal distribution $p(\mathbf{x}_t)$. It can be shown that $\hat{\mathbf{x}}_1$ can also be expressed in terms of the pre-trained velocity field $\boldsymbol{b}_\theta(\mathbf{x}_t, t)$ as follows,

$$\hat{\mathbf{x}}_1 = \frac{1}{\gamma_t \dot{\alpha}_t - \dot{\gamma}_t \alpha_t} \Big[ -\dot{\gamma}_t \mathbf{x}_t + \gamma_t \boldsymbol{b}_\theta(\mathbf{x}_t, t) \Big] \tag{97}$$

## B.2 Conditional Conjugate Integrators for Flows

Analogous to diffusion models, we can design conditional conjugate samplers for flows that treat the conditional score $\nabla_{\mathbf{x}_t} \log p(\mathbf{y}|\mathbf{x}_t)$ as a black box. Similar to Proposition 1, by introducing the transformation $\bar{\mathbf{x}}_t = \boldsymbol{A}_t \mathbf{x}_t$, we have the projected flow dynamics,

$$d\hat{\mathbf{x}}_t = \boldsymbol{A}_t \boldsymbol{B}_t \boldsymbol{A}_t^{-1} \hat{\mathbf{x}}_t dt + d\boldsymbol{\Phi}_t \boldsymbol{b}_\theta(\mathbf{x}_t, t) + w_t r_t^{-2} \frac{\partial \hat{\mathbf{x}}_1}{\partial \mathbf{x}_t}^\top (\boldsymbol{H}^\dagger \boldsymbol{y} - \boldsymbol{P}\hat{\mathbf{x}}_1) dt \tag{98}$$

$$\boldsymbol{A}_t = \exp\left( \int_0^t \boldsymbol{B}_s ds \right), \qquad \boldsymbol{\Phi}_t = \int_0^t \boldsymbol{A}_s ds, \tag{99}$$

where $\boldsymbol{H}^\dagger = \boldsymbol{H}^\top(\boldsymbol{H}\boldsymbol{H}^\top)^{-1}$ and $\boldsymbol{P} = \boldsymbol{H}^\top(\boldsymbol{H}\boldsymbol{H}^\top)^{-1}\boldsymbol{H}$ represent the pseudoinverse and the orthogonal projector operators for the degradation operator $\boldsymbol{H}$. For $\boldsymbol{B}_t = 0$, the formulation in Eqn. 98 becomes equivalent to the $\Pi$GDM formulation proposed for OT-flows in Pokle et al. [2024]. For simplicity, since in this work, we only explore the parameterization in Eqn. 98 for $\boldsymbol{B}_t = 0$, we refer to this parameterization as $\Pi$*GFM*.

### B.2.1 Conjugate-$\Pi$GFM (C-$\Pi$GFM)

Analogous to the discussion of C-$\Pi$GDM samplers in Section 2.2. More specifically, given a noiseless linear inverse problem with $\sigma_y = 0$, and the conditional score $\nabla_{\mathbf{x}_t} \log p(\mathbf{y}|\mathbf{x}_t)$, introducing the transformation $\bar{\mathbf{x}}_t = \boldsymbol{A}_t \mathbf{x}_t$, where

$$\boldsymbol{A}_t = \exp\left[ \int_0^t \boldsymbol{B}_s + \frac{w_s r_s^{-2} \gamma_t \dot{\gamma}_t^2}{2\alpha_t \left( \gamma_t \dot{\alpha}_t - \dot{\gamma}_t \alpha_t \right)} \boldsymbol{P} ds \right] \tag{100}$$

induces the following projected flow dynamics.

$$d\bar{\mathbf{x}}_t = \boldsymbol{A}_t \boldsymbol{B}_t \boldsymbol{A}_t^{-1} \bar{\mathbf{x}}_t dt + d\boldsymbol{\Phi}_y \boldsymbol{y} + d\boldsymbol{\Phi}_b \boldsymbol{b}_\theta(\mathbf{x}_t, t) + d\boldsymbol{\Phi}_j \Big[ \partial_{\mathbf{x}_t} \boldsymbol{b}_\theta(\mathbf{x}_t, t)(\boldsymbol{H}^\dagger \boldsymbol{y} - \boldsymbol{P}\hat{\mathbf{x}}_1) \Big] \tag{101}$$

where,

$$\boldsymbol{\Phi}_y = - \int_0^t \frac{w_s r_t^{-2} \gamma_s \dot{\gamma}_s}{\alpha_s} \boldsymbol{A}_s \boldsymbol{H}^\dagger ds \tag{102}$$

---

**Algorithm 2** *Conjugate ΠGFM sampling*

---

1: **Input:** Corrupted observation $y$, Corruption operator $\boldsymbol{H}$, Pretrained Flow $\boldsymbol{b_\theta}(.,.)$, Choice of $\boldsymbol{B}_t$, NFE budget $N$, Timestep discretization $\{t_i\}_{i=0}^N$, Flow kernel $\mathbf{x}_t = \alpha_t \mathbf{x}_1 + \gamma_t \mathbf{z}$, Start time $\tau$.
2: **Output:** Clean sample $\hat{\mathbf{x}}_1$

3: Pre-Compute $\{\boldsymbol{A}_{t_i}\}_{i=0}^N$ (Eqn. 100)   ▷ Pre-compute coefficients
4: Pre-Compute $\{\boldsymbol{\Phi}_y^i, \boldsymbol{\Phi}_b^i, \boldsymbol{\Phi}_j^i\}_{i=0}^N$ (see Eqns. 102-104)

5: $\mathbf{z} \sim \mathcal{N}(0, \boldsymbol{I}_d)$   ▷ Draw initial samples from the generative prior
6: $\mathbf{x} = \alpha_\tau \boldsymbol{H}^\dagger y + \gamma_\tau \mathbf{z}$   ▷ Initialize using the pseudoinverse (See Chung et al. [2022b])
7: $\bar{\mathbf{x}} = \boldsymbol{A}_\tau \mathbf{x}$   ▷ Initial Projection Step

8: **for** $n = 0$ **to** $N - 1$ **do**
9:     $h = (t_{n+1} - t_n)$   ▷ Time step differential
10:     $\mathbf{x} = \boldsymbol{A}_{t_n}^{-1}\bar{\mathbf{x}}$
11:     $\hat{\mathbf{x}}_1 = \frac{1}{\gamma_t \dot{\alpha}_t - \dot{\gamma}_t \alpha_t}\left[-\dot{\gamma}_t \mathbf{x}_t + \gamma_t \boldsymbol{b_\theta}(\mathbf{x}_t, t)\right]$   ▷ Tweedie's Estimate
12:     $\boldsymbol{v}_l = h\boldsymbol{A}_{t_n}\boldsymbol{B}_{t_n}\boldsymbol{A}_{t_n}^{-1}\bar{\mathbf{x}} + (\boldsymbol{\Phi}_y^{n+1} - \boldsymbol{\Phi}_y^n)y$   ▷ Linear drift
13:     $\boldsymbol{v}_{nl} = (\boldsymbol{\Phi}_b^{n+1} - \boldsymbol{\Phi}_b^n)\boldsymbol{b_\theta}(\mathbf{x}, t_n) + (\boldsymbol{\Phi}_j^{n+1} - \boldsymbol{\Phi}_j^n)\left[\partial_\mathbf{x}\boldsymbol{b_\theta}(\mathbf{x}, t_n)(\boldsymbol{H}^\dagger y - \boldsymbol{P}\hat{\mathbf{x}}_1)\right]$   ▷ Non-Linear drift
14:     $\bar{\mathbf{x}} = \bar{\mathbf{x}} + \boldsymbol{v}_l + \boldsymbol{v}_{nl}$   ▷ Euler Update
15: **end for**

   **return** $\mathbf{x} = \boldsymbol{A}_{t_N}^{-1}\bar{\mathbf{x}}$   ▷ Project back to original space when done

---

$$\boldsymbol{\Phi}_b = \int_0^t \boldsymbol{A}_s\left[\boldsymbol{I}_d + \frac{w_s r_s^{-2}\gamma_s^2\dot{\gamma}_s}{\alpha_s(\gamma_s\dot{\alpha}_s - \dot{\gamma}_s\alpha_s)}\boldsymbol{P}\right]ds \tag{103}$$

$$\boldsymbol{\Phi}_j = \int_0^t \frac{w_s r_s^{-2}\gamma_s^2}{\alpha_s}\boldsymbol{A}_s ds \tag{104}$$

where $\exp(.)$ denotes the matrix exponential, $\boldsymbol{H}^\dagger$, and $\boldsymbol{P}$ are the pseudoinverse and projector operators (as defined previously). Lastly, the matrix $\boldsymbol{B}_t$ is a design choice of our method. We specify a recipe for C-ΠGFM sampling in Algorithm 2.

## C   Extension to Noisy and Non-linear Inverse Problems

Here, we discuss an extension of Conditional Conjugate Integrators to noisy and non-linear inverse problems. While our discussion is primarily in the context of diffusion models, similar theoretical arguments also apply to Flows.

**Noisy Linear Inverse Problems:** For noisy linear inverse problems of the form,

$$\mathbf{y} = \boldsymbol{H}\mathbf{x}_0 + \sigma_y\mathbf{z}, \tag{105}$$

for VPSDE diffusion, the *noisy* transformation $\boldsymbol{A}_t^{\sigma_y}$ can be approximated from the transformation $\boldsymbol{A}_t$ for the noiseless case (i.e., $\sigma_y = 0$) as illustrated in the following result (Proof in Appendix A.5):

$$\boldsymbol{A}_t^{\sigma_y} = \boldsymbol{A}_t + \kappa_3(t)\boldsymbol{H}^\dagger(\boldsymbol{H}^\dagger)^\top + \mathcal{O}(\sigma_y^4) \approx \boldsymbol{A}_t + \kappa_3(t)\boldsymbol{H}^\dagger(\boldsymbol{H}^\dagger)^\top, \tag{106}$$

$$\kappa_3(t) = \frac{w\sigma_y^2}{2}\left(\int_0^t \frac{\beta_s}{r_s^2}ds\right)\left[\exp\left(\kappa_1(t) + \kappa_2(t)\right) - 1\right]. \tag{107}$$

Consequently, the inverse projection $(\boldsymbol{A}_t^{\sigma_y})^{-1}$ can be approximated from $\boldsymbol{A}_t^{\sigma_y}$ from perturbation analysis.

$$(\boldsymbol{A}_t^{\sigma_y})^{-1} \approx \boldsymbol{A}_t^{-1} - \kappa_3(t)\boldsymbol{A}_t^{-1}\boldsymbol{H}^\dagger(\boldsymbol{H}^\dagger)^\top\boldsymbol{A}_t^{-1} + \mathcal{O}(\sigma_y^4) \tag{108}$$

Therefore, the transformation matrix $\boldsymbol{A}_t^{\sigma_y}$ and its inverse (see Appendix A.5) can also be computed tractably for the noisy case. Since, for most practical purposes, $\sigma_y$ is pretty small, higher order terms in $\sigma_y^4$ can be safely ignored, making our approximation accurate. We include qualitative examples for 4x super-resolution with $\sigma_y = 0.05$ for the ImageNet dataset in Figure 8

**Non-Linear Inverse Problems:** For non-linear inverse problems of the form,

$$\boldsymbol{y} = h(\mathbf{x}_0) + \sigma_y \mathbf{z}, \quad \mathbf{z} \sim \mathcal{N}(0, \boldsymbol{I}_d), \tag{109}$$

similar to Song et al. [2022], we heuristically re-define linear operations like $\boldsymbol{H}^\dagger \mathbf{x}_t$, $\boldsymbol{H} \mathbf{x}_t$ and $\boldsymbol{P} \mathbf{x}_t$ by their non-linear equivalents $h^\dagger(\mathbf{x}_t)$, $h(\mathbf{x}_t)$ and $h^\dagger(h(\mathbf{x}_t))$ respectively. Consequently, analogous to Eqn. 11 the projection operator for a noiseless non-linear inverse problem, in this case, can be defined as,

$$A_t = \exp(\kappa_1(t))\Big[\boldsymbol{I}_d + (\exp(\kappa_2(t)) - 1)P\Big], \quad \kappa_1(t) = \int_0^t \Big(\lambda + \frac{1}{2}\beta_s\Big) ds, \quad \kappa_2(t) = -\frac{w}{2}\int_0^t \beta_s ds, \tag{110}$$

where $P = h^\dagger(h(.))$ is non-linear 'projector" operator. For instance, in non-linear inverse problems like compression artifact removal, $h(\mathbf{x}_t)$ and $h^\dagger(\mathbf{x}_t)$ can realized by encoders and decoders. We illustrate some qualitative examples in Figure 12. It is worth noting that this is a purely heuristic approximation, and developing a more principled framework for non-linear inverse problems within our framework remains an interesting direction for further work.

# D    Implementation Details

In this section, we include additional practical implementation details for both C-ΠGDM and C-ΠGFM formulations.

## D.1    C-ΠGDM: Practical Aspects

### D.1.1    VP-SDE

We work with the VP-SDE diffusion [Song et al., 2020] with the forward process specified as:

$$d\mathbf{x}_t = -\frac{1}{2}\beta_t \mathbf{x}_t \, dt + \sqrt{\beta_t} \, d\mathbf{w}_t, \quad t \in [0, T], \tag{111}$$

This implies, $\boldsymbol{F}_t = -\frac{1}{2}\beta_t$ and $\boldsymbol{G}_t = \sqrt{\beta_t}$. For the VP-SDE the perturbation kernel is given by,

$$p(\mathbf{x}_t | \mathbf{x}_0) = \mathcal{N}(\mu_t \mathbf{x}_0, \sigma_t^2 \boldsymbol{I}_d) \tag{112}$$

$$\mu_t = \exp\Big(-\frac{1}{2}\int_0^s \beta_s ds\Big) \qquad \sigma_t^2 = \Big[1 - \exp\Big(-\int_0^s \beta_s ds\Big)\Big] \tag{113}$$

The corresponding deterministic reverse process is parameterized as:

$$d\mathbf{x}_t = -\frac{\beta_t}{2}\Big[\mathbf{x}_t + \boldsymbol{s}_\theta(\mathbf{x}_t, t)\Big] dt. \tag{114}$$

Moreover, we adopt the standard $\epsilon$-prediction parameterization which implies $\boldsymbol{C}_{\text{out}}(t) = -1/\sigma_t$. Lastly, the Tweedies estimate $\hat{\mathbf{x}}_0$ can be specified as:

$$\hat{\mathbf{x}}_0 = \frac{1}{\mu_t}\Big[\mathbf{x}_t + \sigma_t^2 \boldsymbol{s}_\theta(\mathbf{x}_t, t)\Big] \tag{115}$$

### D.1.2    C-ΠGDM - Simplified Expressions

We choose the parameterization $\boldsymbol{B}_t = \lambda \boldsymbol{I}_d$ and set the adaptive guidance weight as $w_t = w\mu_t^2 r_t^2$, where $r_t^2 = \frac{\sigma_t^2}{\sigma_t^2 + \mu_t^2}$. The projected diffusion dynamics are then specified as:

$$d\bar{\mathbf{x}}_t = \lambda\bar{\mathbf{x}}_t dt + d\boldsymbol{\Phi}_y \boldsymbol{y} + d\boldsymbol{\Phi}_s \boldsymbol{\epsilon}_\theta(\mathbf{x}_t, t) + d\boldsymbol{\Phi}_j\Big[\partial_{\mathbf{x}_t}\boldsymbol{\epsilon}_\theta(\mathbf{x}_t, t)(\boldsymbol{H}^\dagger \boldsymbol{y} - \boldsymbol{P}\hat{\mathbf{x}}_0)\Big] \tag{116}$$

where

$$\boldsymbol{A}_t = \exp\Big[\int_0^t \Big(\lambda + \frac{1}{2}\beta_s\Big) ds \boldsymbol{I}_d - \frac{w}{2}\Big(\int_0^t \beta_s ds\Big)\boldsymbol{P}\Big] \tag{117}$$

which further simplifies to,

$$\boldsymbol{A}_t = \exp(\kappa_1(t))\Big[\boldsymbol{I}_d + (\exp(\kappa_2(t)) - 1)\boldsymbol{P}\Big], \quad \kappa_1(t) = \int_0^t \Big(\lambda + \frac{1}{2}\beta_s\Big) ds, \quad \kappa_2(t) = -\frac{w}{2}\int_0^t \beta_s ds \tag{118}$$

Moreover, we have,

$$\boldsymbol{\Phi}_y = -\int_0^t \frac{w_s r_s^{-2}}{2\mu_s} \boldsymbol{A}_s \boldsymbol{G}_s \boldsymbol{G}_s^\top \boldsymbol{H}^\dagger ds \tag{119}$$

$$= -\int_0^t \frac{w\beta_t \mu_s}{2} \boldsymbol{A}_s \boldsymbol{H}^\dagger ds \tag{120}$$

$$= -\int_0^t \frac{w\beta_t \mu_s}{2} \Big[ \exp(\kappa_1(s)) \Big[ \boldsymbol{I}_d + (\exp(\kappa_2(s)) - 1) \boldsymbol{P} \Big] \Big] \boldsymbol{H}^\dagger ds \tag{121}$$

$$= -\int_0^t \frac{w\beta_t \mu_s}{2} \exp(\kappa_1(s)) \Big[ \boldsymbol{H}^\dagger + (\exp(\kappa_2(s)) - 1) \boldsymbol{P} \boldsymbol{H}^\dagger \Big] ds \tag{122}$$

$$= -\int_0^t \frac{w\beta_t \mu_s}{2} \exp(\kappa_1(s)) \Big[ \boldsymbol{H}^\dagger + (\exp(\kappa_2(s)) - 1) \boldsymbol{H}^\dagger \Big] ds \tag{123}$$

$$= -\Big[ \int_0^t \frac{w\beta_t \mu_s}{2} \exp(\kappa_1(s) + \kappa_2(s)) ds \Big] \boldsymbol{H}^\dagger \tag{124}$$

$$\boldsymbol{\Phi}_s = -\int_0^t \frac{1}{2} \boldsymbol{A}_s \boldsymbol{G}_s \boldsymbol{G}_s^\top \Big[ \boldsymbol{I}_d - \frac{w_s r_s^{-2} \sigma_s^2}{\mu_s^2} \boldsymbol{P} \Big] \boldsymbol{C}_{\text{out}}(s) ds \tag{125}$$

$$= \int_0^t \frac{\beta_s}{2\sigma_s} \boldsymbol{A}_s \Big[ \boldsymbol{I}_d - w\sigma_s^2 \boldsymbol{P} \Big] ds \tag{126}$$

$$= \int_0^t \frac{\beta_s}{2\sigma_s} \boldsymbol{A}_s ds - \Big[ \int_0^t \frac{w\beta_s \sigma_s}{2} \exp(\kappa_1(s) + \kappa_2(s)) ds \Big] \boldsymbol{P} \tag{127}$$

$$= \int_0^t \frac{\beta_s}{2\sigma_s} \exp(\kappa_1(s)) \Big[ \boldsymbol{I}_d + (\exp(\kappa_2(s)) - 1) \boldsymbol{P} \Big] ds - \Big[ \int_0^t \frac{w\beta_s \sigma_s}{2} \exp(\kappa_1(s) + \kappa_2(s)) ds \Big] \boldsymbol{P} \tag{128}$$

$$= \int_0^t \frac{\beta_s}{2\sigma_s} \exp(\kappa_1(s)) ds + \Big[ \int_0^t \frac{\beta_s}{2\sigma_s} \exp(\kappa_1(s))(\exp(\kappa_2(s)) - 1) - \frac{w\beta_s \sigma_s}{2} \exp(\kappa_1(s) + \kappa_2(s)) ds \Big] \boldsymbol{P} \tag{129}$$

$$\boldsymbol{\Phi}_j = -\int_0^t \frac{w_s r_s^{-2} \sigma_s^2}{2\mu_s} \boldsymbol{A}_s \boldsymbol{G}_s \boldsymbol{G}_s^\top \boldsymbol{C}_{\text{out}}(s) ds = \int_0^t \frac{w\beta_s \mu_s \sigma_s}{2} \boldsymbol{A}_s ds \tag{130}$$

$$= \int_0^t \frac{w\beta_s \mu_s \sigma_s}{2} \exp(\kappa_1(s)) \Big[ \boldsymbol{I}_d + (\exp(\kappa_2(s)) - 1) \boldsymbol{P} \Big] ds \tag{131}$$

$$= \int_0^t \frac{w\beta_s \mu_s \sigma_s}{2} \exp(\kappa_1(s)) ds + \Big[ \int_0^t \frac{w\beta_s \mu_s \sigma_s}{2} \exp(\kappa_1(s))(\exp(\kappa_2(s)) - 1) ds \Big] \boldsymbol{P} \tag{132}$$

## D.2 C-ΠGFM: Practical Aspects

### D.2.1 OT-Flows

We work with OT-Flows [Albergo et al., 2023, Lipman et al., 2023, Liu et al., 20223] due to its wide adoption. More specifically, the corresponding interpolant can be specified as,

$$\mathbf{x}_t = (1 - t)\mathbf{z} + t\mathbf{x}_1, \quad \mathbf{z} \sim \mathcal{N}(0, \boldsymbol{I}_d) \quad \mathbf{x}_1 \sim p_{\text{data}} \tag{133}$$

For this case $\alpha_t = t$ and $\gamma_t = 1 - t$. Therefore, the Tweedie's estimate of $\mathbb{E}(\mathbf{x}_t | \mathbf{x}_1)$ can be specified as (from Eqn. 97):

$$\hat{\mathbf{x}}_1 = \mathbf{x}_t + (1 - t)\boldsymbol{b}_\theta(\mathbf{x}_t, t) \tag{134}$$

### D.2.2 C-ΠGFM: Simplified Expressions

We choose the parameterization $\boldsymbol{B}_t = \lambda \boldsymbol{I}_d$ and set the adaptive guidance weight as $w_t = w\alpha_t^2 r_t^2$, where $r_t^2 = \frac{\gamma_t^2}{\alpha_t^2 + \gamma_t^2}$. The projected diffusion dynamics are then specified as follows:

$$d\bar{\mathbf{x}}_t = \lambda\bar{\mathbf{x}}_t dt + d\boldsymbol{\Phi}_y \boldsymbol{y} + d\boldsymbol{\Phi}_b \boldsymbol{b}_\theta(\mathbf{x}_t, t) + d\boldsymbol{\Phi}_j \Big[ \partial_{\mathbf{x}_t} \boldsymbol{b}_\theta(\mathbf{x}_t, t)(\boldsymbol{H}^\dagger \boldsymbol{y} - \boldsymbol{P}\hat{\mathbf{x}}_1) \Big] \tag{135}$$

where,

$$\boldsymbol{A}_t = \mathbf{exp}\Big[\int_0^t \lambda \boldsymbol{I}_d + \frac{wt(1-t)}{2}\boldsymbol{P}ds\Big] \tag{136}$$

which further simplifies to,

$$\boldsymbol{A}_t = \exp(\kappa_1(t))\Big[\boldsymbol{I}_d + (\exp(\kappa_2(t))-1)\boldsymbol{P}\Big], \quad \kappa_1(t) = \int_0^t \lambda ds, \quad \kappa_2(t) = \frac{w}{2}\int_0^t s(1-s)ds \tag{137}$$

Moreover, we have,

$$\boldsymbol{\Phi}_y = -\int_0^t \frac{w_s r_t^{-2}\gamma_s \dot{\gamma}_s}{\alpha_s}\boldsymbol{A}_s \boldsymbol{H}^\dagger ds \tag{138}$$

$$= -\int_0^t w\alpha_s\gamma_s\dot{\gamma}_s\boldsymbol{A}_s\boldsymbol{H}^\dagger ds = \int_0^t ws(1-s)\boldsymbol{A}_s\boldsymbol{H}^\dagger ds \tag{139}$$

$$= \int_0^t ws(1-s)\exp(\kappa_1(s))\Big[\boldsymbol{I}_d + (\exp(\kappa_2(s))-1)\boldsymbol{P}\Big]\boldsymbol{H}^\dagger ds \tag{140}$$

$$= \Big[\int_0^t ws(1-s)\exp(\kappa_1(s)+\kappa_2(s))ds\Big]\boldsymbol{H}^\dagger \tag{141}$$

$$\boldsymbol{\Phi}_b = \int_0^t \boldsymbol{A}_s\Big[\boldsymbol{I}_d + \frac{w_s r_s^{-2}\gamma_s^2\dot{\gamma}_s}{\alpha_s(\gamma_s\dot{\alpha}_s - \dot{\gamma}_s\alpha_s)}\boldsymbol{P}\Big]ds \tag{142}$$

$$= \int_0^t \boldsymbol{A}_s\Big[\boldsymbol{I}_d + \frac{w\alpha_s\gamma_s^2\dot{\gamma}_s}{(\gamma_s\dot{\alpha}_s - \dot{\gamma}_s\alpha_s)}\boldsymbol{P}\Big]ds \tag{143}$$

$$= \int_0^t \boldsymbol{A}_s\Big[\boldsymbol{I}_d - ws(1-s)^2\boldsymbol{P}\Big]ds \tag{144}$$

$$= \int_0^t \boldsymbol{A}_s ds - \int_0^t ws(1-s)^2\boldsymbol{A}_s\boldsymbol{P}\Big]ds \tag{145}$$

$$= \int_0^t \boldsymbol{A}_s ds - \Big[\int_0^t ws(1-s)^2\exp(\kappa_1(s)+\kappa_2(s))ds\Big]\boldsymbol{P} \tag{146}$$

$$= \int_0^t \exp(\kappa_1(s))ds + \Big[\int_0^t \exp(\kappa_1(s))(\exp(\kappa_2(s))-1) - ws(1-s)^2\exp(\kappa_1(s)+\kappa_2(s))ds\Big]\boldsymbol{P} \tag{147}$$

$$\boldsymbol{\Phi}_j = \int_0^t \frac{w_s r_s^{-2}\gamma_s^2}{\alpha_s}\boldsymbol{A}_s ds = \int_0^t w\alpha_s\gamma_s^2\boldsymbol{A}_s ds \tag{148}$$

$$= \int_0^t ws(1-s)^2\exp(\kappa_1(s))\Big[\boldsymbol{I}_d + (\exp(\kappa_2(s))-1)\boldsymbol{P}\Big]ds \tag{149}$$

$$= \int_0^t ws(1-s)^2\exp(\kappa_1(s))ds + \Big[\int_0^t ws(1-s)^2\exp(\kappa_1(s))(\exp(\kappa_2(s))-1)ds\Big]\boldsymbol{P} \tag{150}$$

### D.3 Coefficient Computation

From the above analysis, most integrals are one-dimensional and can be computed in closed form or numerically with high precision. To clarify, with a predetermined timestep schedule $\{t_i\}$, the coefficients $\Phi$ can be calculated offline just once and then reused across various samples. Therefore, this computation must only be done once offline for each sampling run. For numerical approximation of these integrals, we use the `odeint` method from the `torchdiffeq` package [Chen, 2018] with parameters `atol=1e-5`, `rtol=1e-5` and the RK45 solver [Dormand and Prince, 1980]. We set the initial value $\boldsymbol{\Phi}_{\text{init}} = \boldsymbol{0}$ for all coefficients $\boldsymbol{\Phi}$ as an initial condition for both C-ΠGDM and C-ΠGFM samplers.

### D.4 Choice of Numerical Solver

We use the Euler method to simulate projected diffusion/flow dynamics for simplicity. However, using higher-order numerical solvers within our framework is also possible. We leave this exploration to future work.

### D.5 Timestep Selection during Sampling

: We use uniform spacing for timestep discretization during sampling. We hypothesize our sampler can also benefit from more advanced timestep discretization techniques Karras et al. [2022] commonly used for sampling in unconditional diffusion models in the low NFE regime.

### D.6 Last-Step Denoising

It is common to add an Euler-based denoising step from a cutoff $\epsilon$ to zero to optimize for sample quality [Song et al., 2020, Dockhorn et al., 2022, Jolicoeur-Martineau et al., 2021] at the expense of another sampling step. In this work, we do not use last-step denoising for our samplers.

### D.7 Evaluation Metrics

We use the network function evaluations (NFE) to assess sampling efficiency and perceptual metrics KID [Bińkowski et al., 2018], LPIPS [Zhang et al., 2018] and FID [Heusel et al., 2017] to assess sample quality. In practice, we use the `torch-fidelity`[Obukhov et al., 2020] package for computing all FID and KID scores reported in this work. For LPIPS, we use the `torchmetrics` package with `Alexnet` embedding.

### D.8 Baseline Hyperparameters

**Diffusion Baselines:** For DPS [Chung et al., 2022a], we set NFE=1000 and set the step size for each task to the value recommended in Appendix D in Chung et al. [2022a]. For DDRM [Kawar et al., 2022], we set the number of sampling steps to NFE=20 with parameters $\eta_b = 1.0$ and $\eta = 0.85$ as recommended in Kawar et al. [2022]. For both DPS and DDRM we start diffusion sampling from $t = T$. For our implementation of $\Pi$-GDM, we set the start time parameter $\tau$ to 0.6 for super-resolution and deblurring. We set the guidance weight $w_t = wr_t^2$ where $w$ is tuned using grid search between 1.0 and 10.0 for best sample quality for super-resolution and deblurring. For implementation of all diffusion-based baselines, we use the official code for RED-Diff [Mardani et al., 2023] at `https://github.com/NVlabs/RED-diff`.

**Flow Baselines:** In developing our flow-based baseline, we adhere to the approach outlined in $\Pi$GFM (Pokle et al., 2024), which advocates for a consistent guidance schedule characterized by $w_t = w$ and $r_t = \frac{\gamma_t^2}{\gamma_t^2 + \alpha_t^2}$. For each task, we perform a comprehensive grid search over the parameters $\alpha_\tau = \{0.1, 0.2, \ldots, 0.7\}$ and $w = \{1, 2, \ldots, 5\}$ (35 combinations in total) across different datasets to identify the optimal configuration that minimizes the LPIPS score. For the implementation of Flows, we use the official implementation of Rectified Flows [Liu et al., 20223] at `https://github.com/gnobitab/RectifiedFlow`.

## E Additional Results

### E.1 Additional Baseline Comparisons

We include additional comparisons between our proposed samplers and competing baselines on the AFHQ-Cat (see Table 2), LSUN Bedroom (see Table 3), and the FFHQ (see Table 4) datasets.

**A note on Inpainting evaluations for ImageNet.** We find that for diffusion model evaluations, the continuous sampler for $\Pi$GDM suffers from noisy artifacts for the inpainting task. Consequently, Conjugate $\Pi$GDM suffers from similar artifacts. Therefore, we do not report results on this task for the ImageNet dataset.

| Task | NFE | LPIPS↓ | | KID×$10^{-3}$↓ | | FID↓ | |
|------|-----|--------|--------|--------|--------|--------|--------|
| | | C-ΠGFM | ΠGFM | C-ΠGFM | ΠGFM | C-ΠGFM | ΠGFM |
| Inpainting | 5 | **0.151** | 0.177 | **6.5** | 15.6 | **21.76** | 30.82 |
| | 10 | **0.122** | 0.136 | **8.5** | 9.4 | **22.50** | 24.87 |
| | 20 | **0.115** | 0.117 | **6.4** | 10.4 | **20.39** | 24.42 |
| Super-Resolution | 5 | **0.129** | 0.133 | **4.1** | 5.7 | **18.43** | 19.55 |
| | 10 | 0.132 | **0.121** | 4.0 | 4.6 | 18.32 | **17.65** |
| | 20 | 0.134 | **0.119** | 4.5 | **4.0** | 18.75 | **16.97** |
| Deblurring | 5 | **0.176** | 0.177 | **6.6** | 6.9 | **23.28** | 23.66 |
| | 10 | 0.182 | **0.164** | 9.4 | **7.1** | 28.12 | **23.62** |
| | 20 | 0.191 | **0.170** | 12.4 | **7.2** | 31.76 | **23.65** |

Table 2: Quantitative evaluation on 4x superresolution, inpainting, and Gaussian deblurring on the AFHQ-Cat dataset.

| Task | NFE | LPIPS↓ | | KID×$10^{-3}$↓ | | FID↓ | |
|------|-----|--------|--------|--------|--------|--------|--------|
| | | C-ΠGFM | ΠGFM | C-ΠGFM | ΠGFM | C-ΠGFM | ΠGFM |
| Inpainting | 5 | **0.208** | - | **7.0** | - | **45.66** | - |
| | 10 | **0.176** | - | **4.4** | - | **40.69** | - |
| | 20 | **0.167** | - | **4.2** | - | **40.35** | - |
| Super-Resolution | 5 | **0.174** | 0.219 | **3.1** | 7.7 | **37.54** | 46.03 |
| | 10 | **0.150** | 0.193 | **1.1** | 4.6 | **32.41** | 37.34 |
| | 20 | **0.148** | 0.175 | **0.9** | 2.5 | 32.26 | **32.15** |
| Deblurring | 5 | **0.209** | 0.220 | **5.0** | 9.0 | **44.78** | 49.27 |
| | 10 | 0.204 | **0.193** | 10.7 | **4.7** | 53.53 | **44.21** |
| | 20 | 0.224 | **0.175** | 18.0 | **3.5** | 62.87 | **39.95** |

Table 3: Quantitative evaluation on 4x superresolution, inpainting, and Gaussian deblurring on the LSUN-Bedroom dataset. We note that ΠGFM fails to generate reasonable texture in the masked region even with the maximum NFE=20, so we choose not to report the results here. (See qualitative examples in Figure 9)

## E.2 Comparison of Perceptual vs Recovery Metrics

Here, we highlight the robustness of C-ΠGDM in both perceptual and recovery metrics in the context of inverse problems. For completeness, we provide a comparison between DPS, ΠGDM, and C-ΠGDM in terms of PSNR, SSIM, FID, and LPIPS in Tables 5 and 6 on the ImageNet-256 and FFHQ-256 datasets on the 4x super-resolution task. It is worth noting that the PSNR and SSIM scores for all methods correspond with the best FID/LPIPS scores presented in the main text for these methods. Our method achieves competitive PSNR and SSIM scores for better perceptual quality than competing baselines like DPS/Π-GDM, even for very small sampling budgets. For instance, on the FFHQ dataset, our method achieves a PSNR of 28.97 compared to 28.49 for DPS while achieving better perceptual sample quality (LPIPS: 0.095 for ours vs 0.107 for DPS) and requiring around 200 times less sampling budget (NFE=5 for our method vs 1000 for DPS). Therefore, we argue that our perceptual quality to recovery trade-off is better than competing baselines.

## E.3 Traversing the Recovery vs Perceptual trade-off

In addition to the guidance weight $w$, our method also allows tuning an additional hyperparameter $\lambda$, which controls the dynamics of the projection operator (See Sections 2.3 and 3.2 for more intuition). Therefore, tuning $w$ and $\lambda$ can help traverse the trade-off curve between perceptual quality and distortion for a fixed NFE budget. We illustrate this aspect in Table 7 (fixed $\lambda$ with varying $w$) and Table 8 (fixed $w$ with varying $\lambda$) for the SR(x4) task on the ImageNet-256 dataset using the PSNR, LPIPS, and FID metrics. Therefore, our method offers greater flexibility to tune the sampling process towards either good perceptual quality or good recovery for a given application while maintaining the same number of sampling steps. In contrast, other methods like DPS or Π-GDM do not offer such

| Task | NFE | LPIPS↓ | | | | KID×$10^{-3}$ ↓ | | | | FID↓ | | | |
|---|---|---|---|---|---|---|---|---|---|---|---|---|---|
| | | C-ΠGDM | ΠGDM | DPS | DDRM | C-ΠGDM | ΠGDM | DPS | DDRM | C-ΠGDM | ΠGDM | DPS | DDRM |
| Super-Resolution | 5 | **0.095** | 0.133 | | | **10.9** | 17.4 | | | **32.01** | 41.39 | | |
| | 10 | **0.086** | 0.106 | 0.106 | 0.106 | **8.8** | 10.2 | 7.8 | 22.8 | **29.07** | 32.79 | 30.86 | 36.95 |
| | 20 | **0.083** | 0.087 | | | 5.8 | **4.6** | | | **26.37** | 26.17 | | |
| Deblurring | 5 | **0.127** | 0.147 | | | **7.3** | 14.6 | | | **31.18** | 39.63 | | |
| | 10 | **0.111** | 0.123 | 0.348 | 0.132 | **6.3** | 7.7 | 109.4 | 11.5 | **29.08** | 31.49 | 142.26 | 33.94 |
| | 20 | 0.112 | **0.103** | | | 4.4 | **3.1** | | | 27.68 | **26.30** | | |

Table 4: Quantitative evaluation on 4x superresolution and Gaussian Deblurring tasks for the FFHQ dataset. DPS was evaluated with NFE=1000 but failed to perform well on the deblurring task. DDRM was evaluated with NFE=20.

| | PSNR ↑ | SSIM ↑ | FID ↓ | LPIPS ↓ |
|---|---|---|---|---|
| DPS (NFE=1000) | **23.81** | **0.708** | 38.18 | 0.252 |
| ΠGDM (NFE=20) | 21.92 | 0.646 | 37.36 | 0.222 |
| C-ΠGDM (NFE=5) | 22.32 | 0.641 | 37.31 | 0.220 |
| C-ΠGDM (NFE=10) | 23.00 | 0.651 | **34.22** | **0.206** |
| C-ΠGDM (NFE=20) | 23.16 | 0.654 | 34.28 | 0.207 |

Table 5: Comparison between C-ΠGDM and other baselines in terms of the Recovery (a.k.a distortion) vs Perception tradeoff for ImageNet-256 dataset for the SR(x4) task.

flexibility. Moreover, tuning the guidance weight in methods like DPS could be very expensive due to its high sampling budget requirement (around 1000 NFE).

### E.4 Qualitative Results

**Diffusion Models:**

1. We include additional qualitative comparisons between Π-GDM and our proposed C-ΠGDM sampler for the ImageNet dataset in Fig. 4.

2. We include a qualitative comparison between sample quality at different sampling budgets for the C-ΠGDM sampler in Fig.5.

3. We qualitatively study the impact of varying $w$ on sample quality in Fig. 6 and the impact of varying $\lambda$ on sample quality in Fig. 7.

4. We qualitatively present the performance of the C-ΠGDM sampler for noisy inverse problems in Fig. 8. In just 5 steps, our method can also generate good-quality samples for noisy inverse problems.

**Flow Models:**

1. We include additional qualitative comparisons between Π-GFM and our proposed C-ΠGFM sampler with different sampling budget for the all three datasets in Fig. 9.

2. We qualitatively study the impact of varying $w$ on sample quality in Fig. 10 and the impact of varying $\lambda$ on sample quality in Fig. 11.

|  | PSNR ↑ | SSIM ↑ | FID ↓ | LPIPS ↓ |
|---|---|---|---|---|
| DPS (NFE=1000) | **28.49** | **0.834** | 30.86 | 0.107 |
| ΠGDM (NFE=20) | 28.26 | 0.818 | **26.17** | 0.087 |
| C-ΠGDM (NFE=5) | 28.97 | 0.832 | 32.01 | 0.095 |
| C-ΠGDM (NFE=10) | 29.03 | 0.821 | 29.07 | 0.086 |
| C-ΠGDM (NFE=20) | 28.79 | 0.809 | 26.37 | **0.083** |

Table 6: Comparison between C-ΠGDM and other baselines in terms of the Recovery (a.k.a distortion) vs Perception tradeoff for FFHQ-256 dataset for the SR(x4) task.

| w | PSNR ↑ | LPIPS ↓ | FID ↓ |
|---|---|---|---|
| 2 | 22.91 | 0.339 | 48.48 |
| 4 | 23.37 | 0.306 | 45.03 |
| 6 | **23.49** | 0.274 | 42.68 |
| 8 | 23.44 | 0.266 | 40.96 |
| 10 | 23.28 | 0.254 | 40.27 |
| 12 | 22.89 | 0.246 | **40.13** |
| 14 | 22.74 | **0.239** | 40.16 |

Table 7: Illustration of the impact of $w$ for a fixed $\lambda = 0.0$ on the sample recovery (PSNR) vs sample perceptual quality (LPIPS, FID) at NFE=5 for our method. The task is SR(x4) on the ImageNet-256 dataset.

| $\lambda$ | PSNR ↑ | LPIPS ↓ | FID ↓ |
|---|---|---|---|
| -1.0 | 20.96 | 0.291 | 42.56 |
| -0.8 | 21.33 | 0.265 | 40.97 |
| -0.6 | 21.69 | 0.240 | 39.38 |
| -0.4 | 22.04 | 0.223 | 37.83 |
| -0.2 | 22.32 | **0.220** | **37.31** |
| 0.2 | 22.73 | 0.257 | 45.27 |
| 0.4 | 22.90 | 0.275 | 48.98 |
| 0.6 | 23.03 | 0.283 | 47.47 |
| 0.8 | 23.11 | 0.285 | 46.2 |
| 1.0 | **23.15** | 0.285 | 46.41 |

Table 8: Illustration of the impact of $\lambda$ for a fixed $w = 15.0$ on the sample recovery (PSNR) vs sample perceptual quality (LPIPS, FID) at NFE=5 for our method. The task is SR(x4) on the ImageNet-256 dataset.

| Original | Pseudoinverse | ΠGDM | C − ΠGDM |
|---|---|---|---|

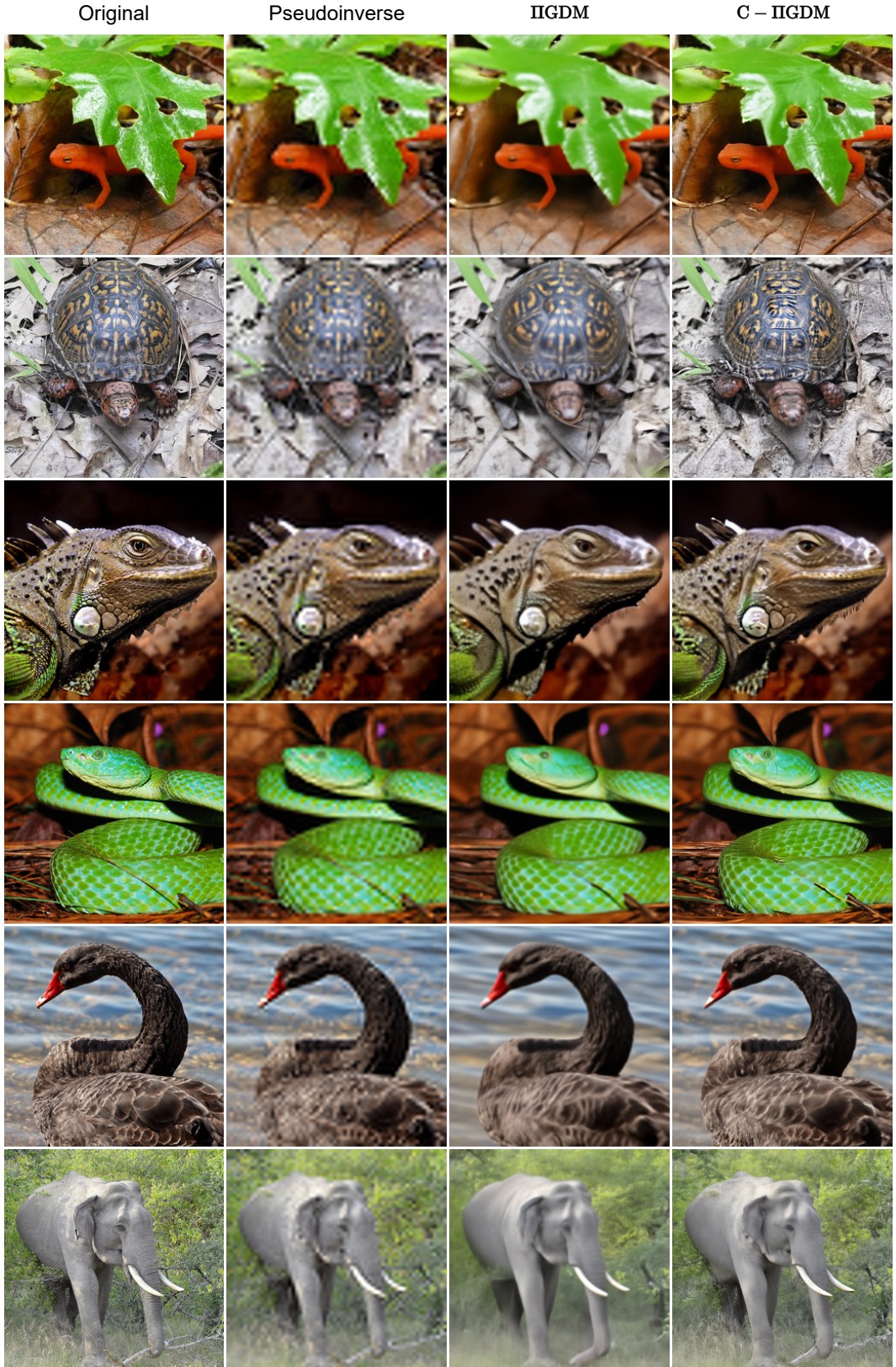

Figure 4: Qualitative comparison between ΠGDM and C-ΠGDM at NFE=5 for the ImageNet dataset on the 4x Superresolution task. C-ΠGDM can generate high-frequency details even for a low compute budget as compared to the baseline Π-GDM (Best Viewed when zoomed in)

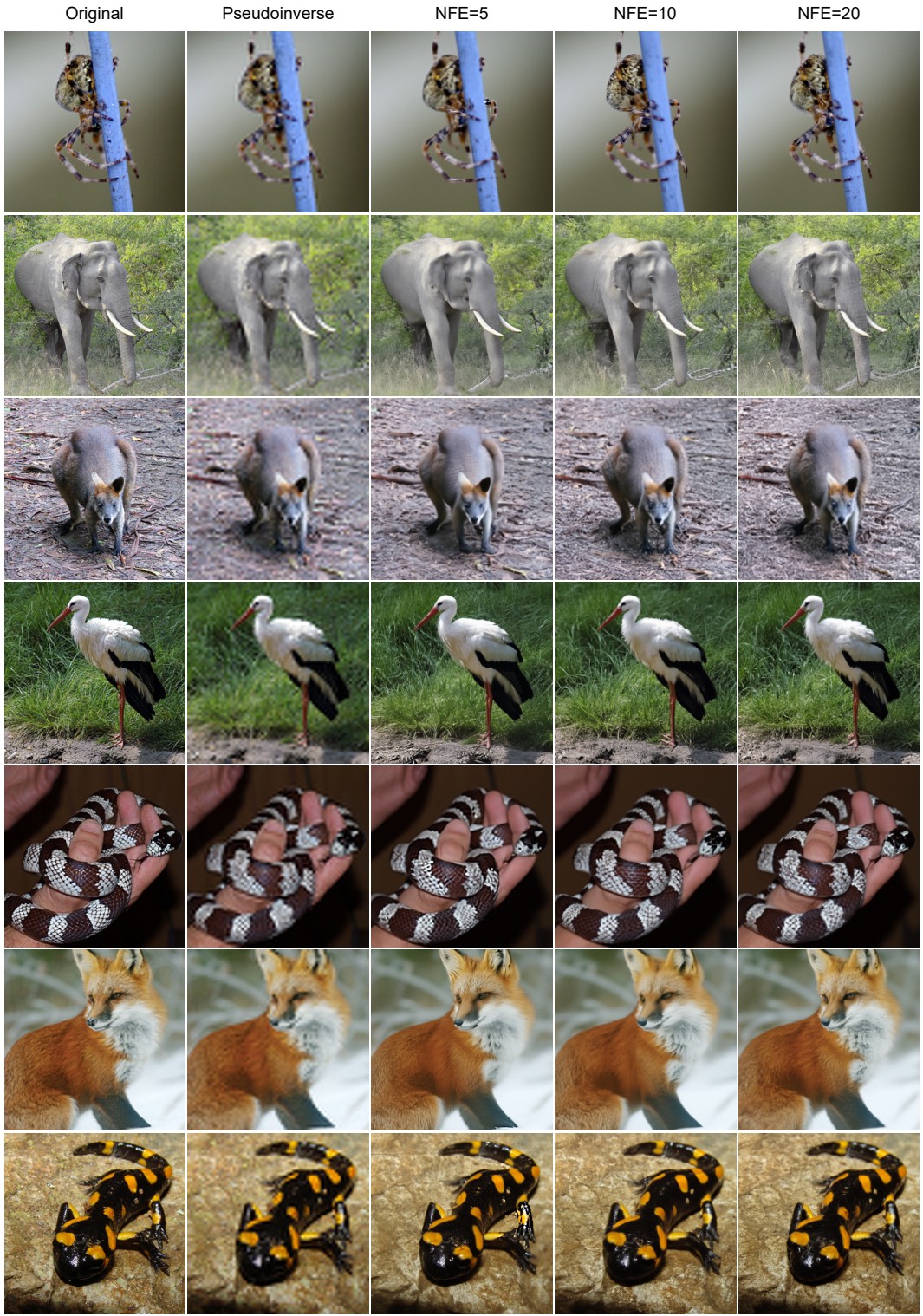

Figure 5: Qualitative comparison for different sampling budgets for the ImageNet dataset on the 4x Superresolution task. C-ΠGDM can generate high-quality samples in just 5 steps (Best Viewed when zoomed in)

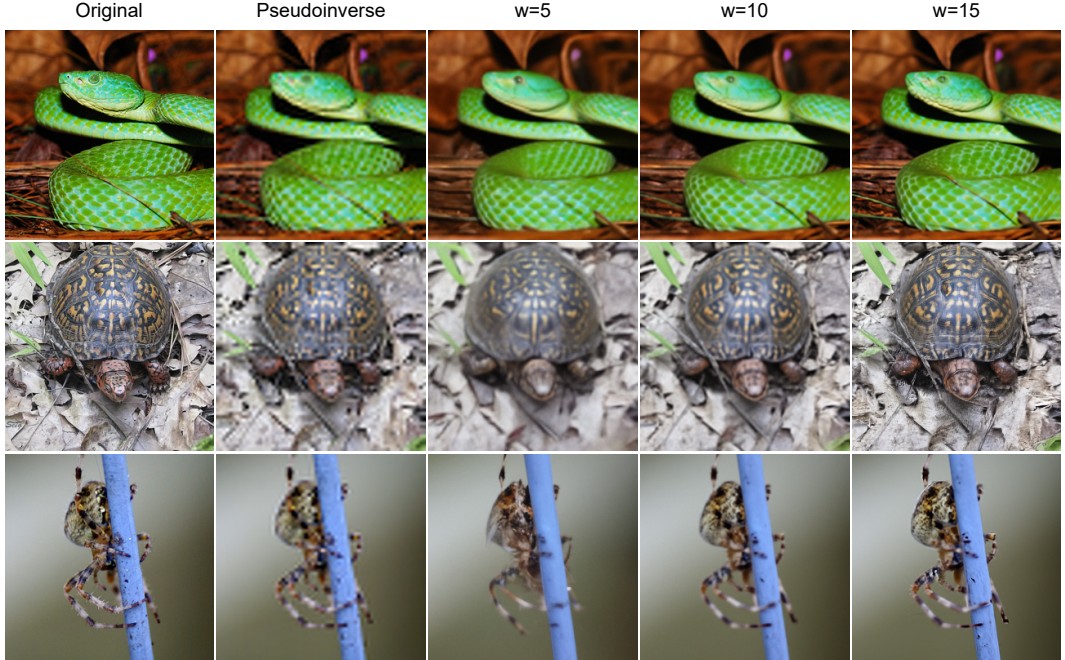

Figure 6: Impact of varying C-ΠGDM guidance weight $w$ on sample quality for the ImageNet dataset on the 4x Superresolution task. High guidance weight is crucial to generate good quality samples from C-ΠGDM (NFE=5 steps) (Best Viewed when zoomed in)

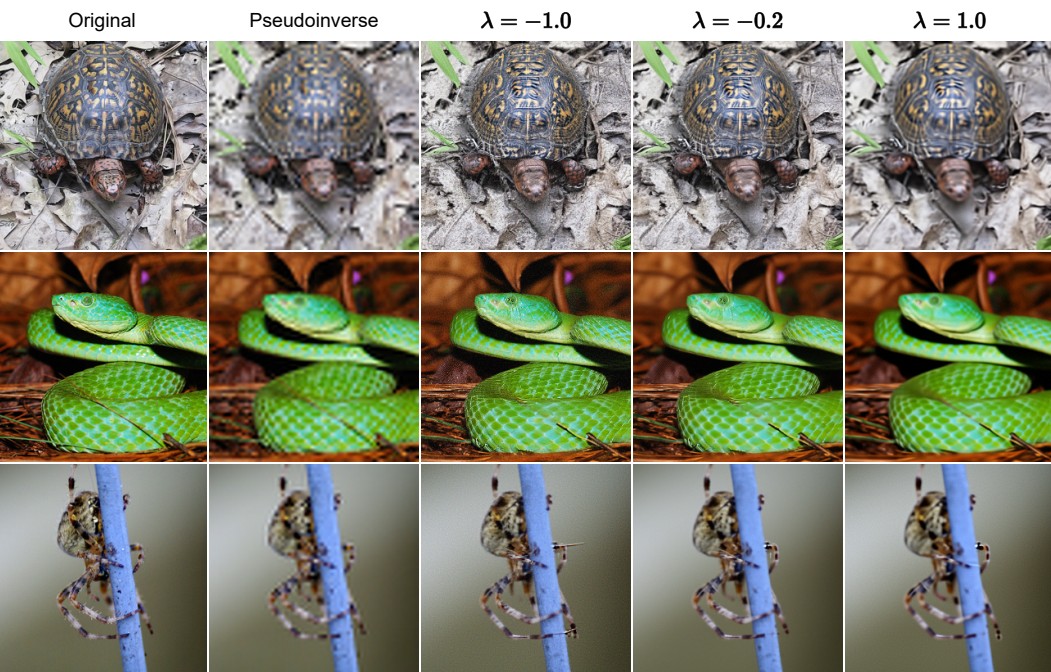

Figure 7: Impact of varying C-ΠGDM $\lambda$ on sample quality for the ImageNet dataset on the 4x Superresolution task. High $\lambda$ can lead to blurry samples while a very low $\lambda$ can lead to over-sharpened artifacts (NFE=5 steps) (Best Viewed when zoomed in)

Original Pseudoinverse C − ΠGDM

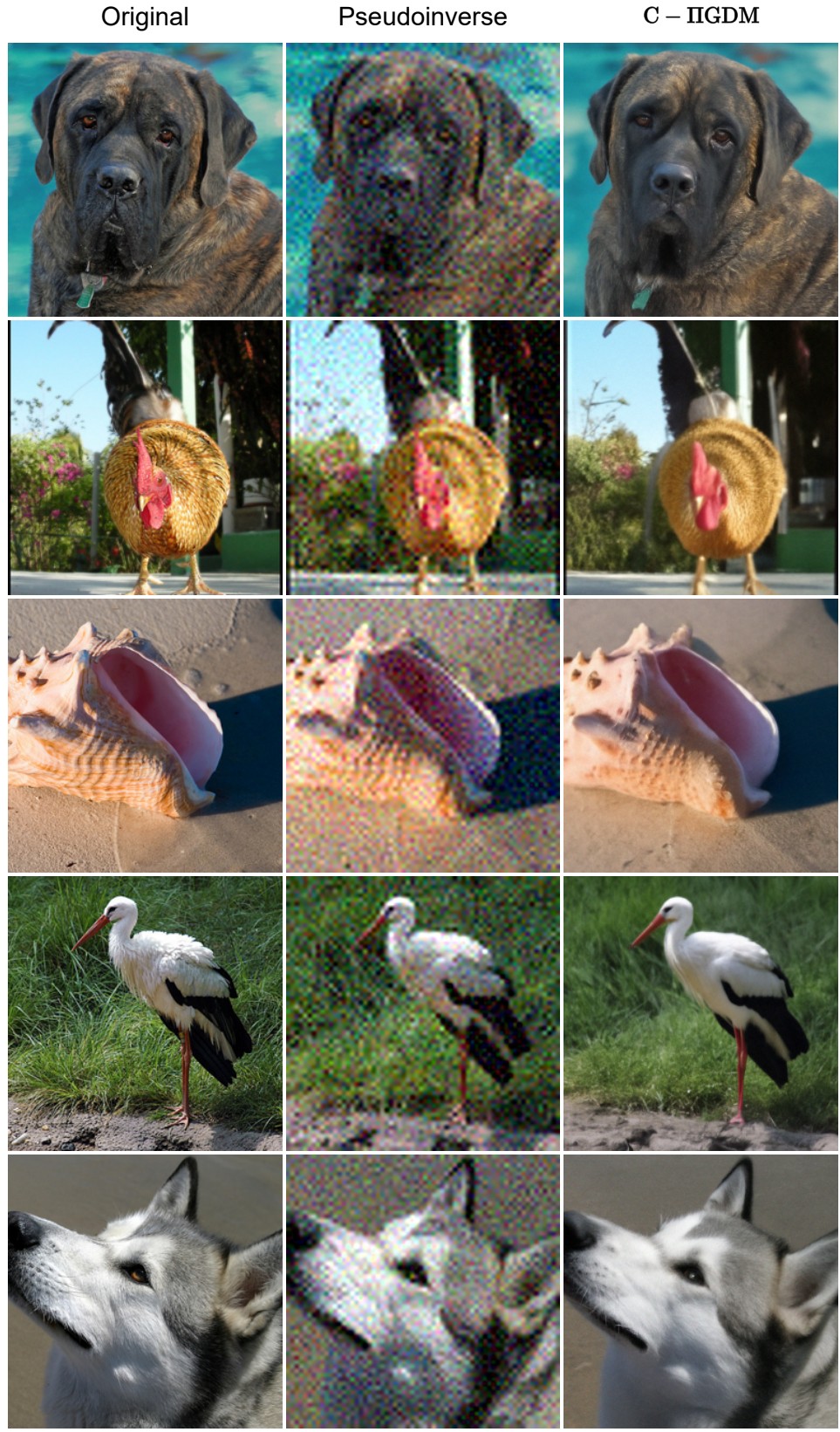

Figure 8: C-ΠGDM can also generate good quality samples for noisy inverse problems (4x superres with NFE=5, $\sigma_y = 0.05$). For this case naively computing the pseudoinverse fails to get rid of the noise.

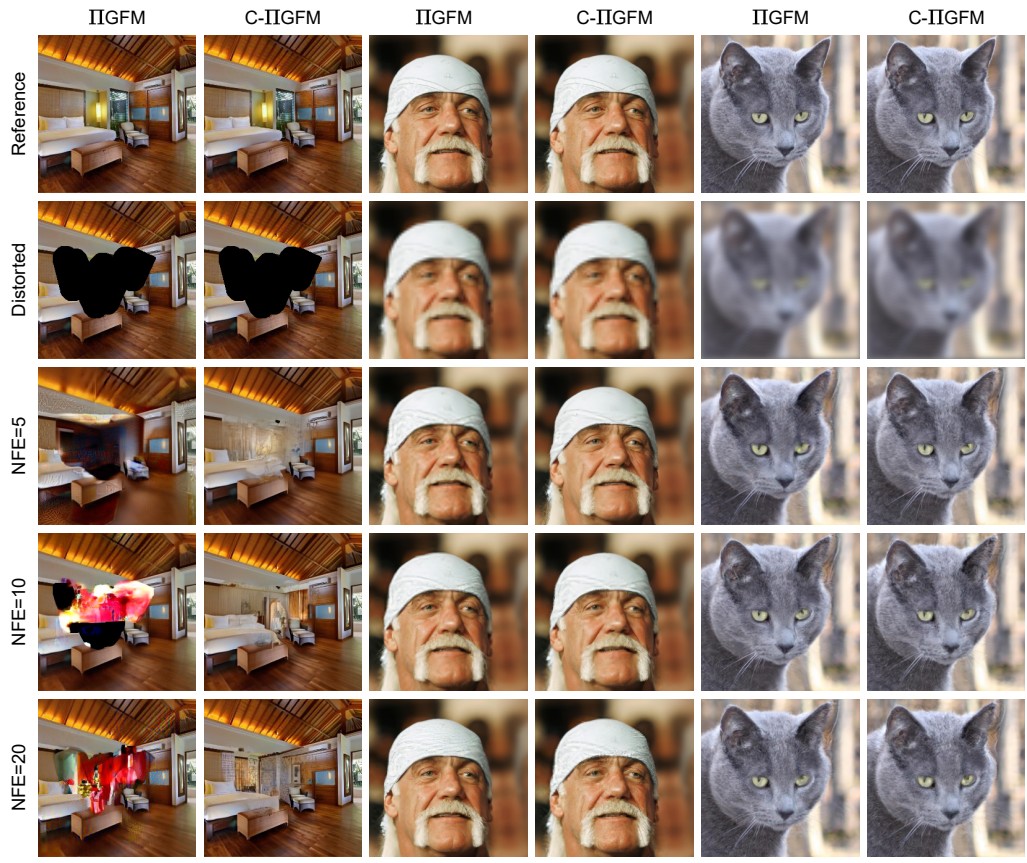

Figure 9: Qualitative comparison between ΠGFM and C-ΠGFM at NFE={5, 10} for the 3 datasets on 3 tasks. C-ΠGFM can generate high-frequency details even for a low compute budget as compared to the baseline Π-GFM (Best Viewed when zoomed in). We did not report ΠGFM inpainting results in Table 3 as it failed to generate "reasonable" textures even after extensive hyper-parameter search on $w$ and $\tau$.

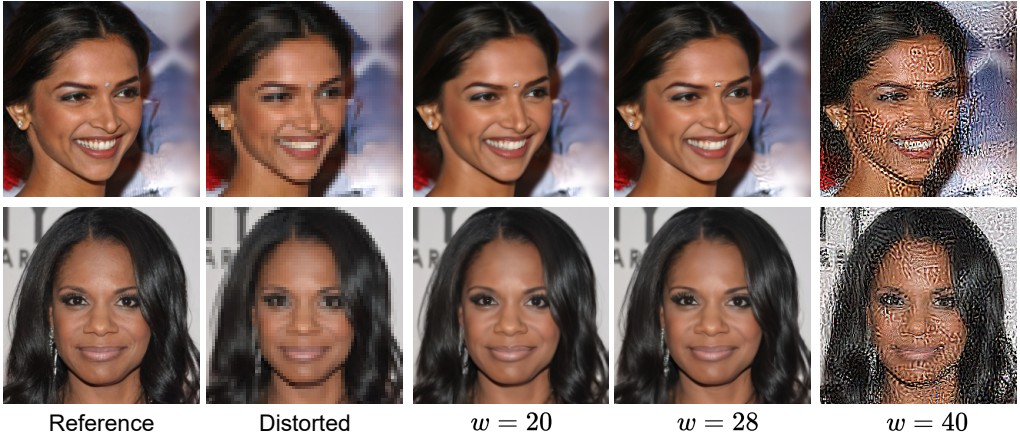

Figure 10: Impact of varying C-ΠGFM guidance weight $w$ on sample quality for the ImageNet dataset on the 4x Superresolution task. High guidance weight is crucial to generate good quality samples from C-ΠGFM (NFE=5 steps) (Best Viewed when zoomed in)

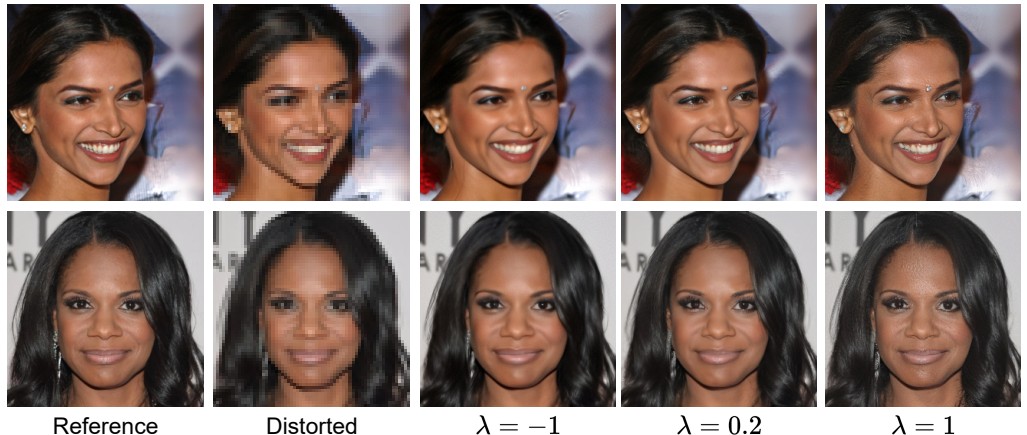

| Reference | Distorted | $\lambda = -1$ | $\lambda = 0.2$ | $\lambda = 1$ |

Figure 11: Impact of varying C-ΠGFM $\lambda$ on sample quality for the ImageNet dataset on the 4x Superresolution task. High $\lambda$ can lead to blurry samples while a very high $\lambda$ can lead to over-sharpened artifacts (NFE=5 steps) (Best Viewed when zoomed in)

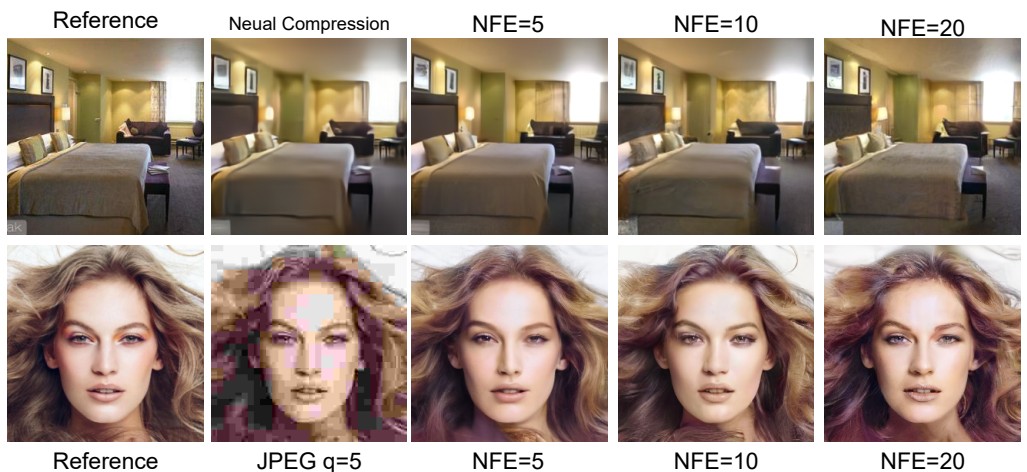

Figure 12: C-ΠGFM for solving compression inverse problem. Top: decoding compressed latents from pretrained mean-scale hyperprior neural codec [Minnen et al., 2018]; Bottom: JPEG image restoration.

