# OpenReview forum: "Fast samplers for Inverse Problems in Iterative Refinement models"
_NeurIPS.cc/2024/Conference — NeurIPS 2024 poster_

### Official Review · Reviewer_X2Z6 · 2024-07-05

**Soundness:** 2
**Presentation:** 2
**Contribution:** 3
**Rating:** 7
**Confidence:** 3

**Summary:**

This paper introduces a inverse problem approach to reconstruct low-resolution blurred or obfuscated images, from pre-trained generative models. The method leverages conjugate integrators to project the diffusion dynamics in an easier space, solve the inverse problem and then map back.

**Strengths:**

- The integration of Conjugate integrators for inverse problems in the paper is rigorously defined and proved.
- The proposed method drastically reduces the number of sampling steps compared to other inverse problem approaches.
- The presented approach achieve better reconstructed images compared with state-of-the-art inverse problem methods.

**Weaknesses:**

The paper is extremely dense with equations and mathematical formulations. I would suggest moving sections 2.3 and 3.4, as well as the numerical details of the experiments, to the appendix. This would allow space in the main paper for additional qualitative results and a broader discussion on the intuition behind the conjugate integrators for inverse problems. Including a figure illustrating the method would also be appreciated. While the mathematical formulation is the main essence of the paper, a visual representation or descriptive insight would be appreciated for the understanding.

**Questions:**

Minor details:
- Line 608, link the approximation to Eq. (5)

**Limitations:**

I don't have any concern.

---

> ### Author Rebuttal · Authors · 2024-08-06
>
> We thank the reviewer for their feedback. We provide a detailed response to several concerns below:
>
> >  I would suggest moving sections 2.3 and 3.4, as well as the numerical details of the experiments, to the appendix.....
>
> We thank the reviewer for their suggestion. We stress that the final version allows for an extra page, which we will use to make the paper more easily accessible. Re. Section 2.3, we would like to highlight that this section provides an intuitive understanding of our method's tractability and working. Therefore, we think this section serves as an intuitive premise for the observations presented in our experimental section.  Re. Section 3.4, we will consider moving this section to the appendix since our presented results for non-linear and noisy inverse problems serve as a proof of concept. Additionally, we plan to add an overview figure in the revised manuscript. We also provide a sample overview figure as Figure 1 in the rebuttal pdf (Also see the second point of our shared response).
>
> >  Line 608, link the approximation to Eq. (5)
>
> Thanks for pointing it out. We will revise our manuscript accordingly

---

### Official Review · Reviewer_PpUR · 2024-07-08

**Soundness:** 3
**Presentation:** 3
**Contribution:** 3
**Rating:** 5
**Confidence:** 3

**Summary:**

Inverse problems like super-resolution and deblurring remain computationally inefficient with current diffusion/flow models. The paper introduces a plug-and-play framework with Conditional Conjugate Integrators that use pre-trained iterative refinement models to simplify sampling. The method generates high-quality samples in as few as 5 steps for tasks like 4× super-resolution on ImageNet, outperforming existing methods.

**Strengths:**

1. The paper is well-written, and the literature review is thorough.
2. The method is well-supported by theoretical foundations.
3. The experiments on both diffusion and flow-matching methods across various tasks demonstrate the robustness of the proposed speed-up algorithm.

**Weaknesses:**

1. The reason why the paper is restricted to the $\Pi$GDM paradigm is unclear.

**Questions:**

1. Can the method be applied to nonlinear inverse problems? If not, please clarify the reason.
2. Though the Conditional Conjugate Integrators work well and can speed up the sampling process to 5 steps, a more detailed computational comparison is needed as each iteration becomes more complicated. Can you provide more detailed computational results?
3. Can you clarify the reason why you restrict yourself to the posterior approximation in $\Pi$GDM? Though the experimental results are promising, a direct comparison with DPS is preferred.

**Limitations:**

Yes, the paper mentioned its broader impact at the end, which looks good to me.

---

> ### Author Rebuttal · Authors · 2024-08-06
>
> $\newcommand\dag\dagger$
> We thank the reviewer for their feedback. We provide a detailed response to several concerns below:
>
> > The reason why the paper is restricted to the $\Pi$GDM paradigm is unclear.
>
> We chose to work with the $\Pi$GDM baseline due to the following reasons:
>
> 1. Firstly, $\Pi$GDM is a very strong and widely-used baseline for tackling inverse problems, providing a more expressive posterior approximation $p(x_0|x_t)$ compared to other methods such as DPS or MCG. This makes it an excellent starting point for low NFE budget sampling. As demonstrated in our experiments (See Table 1), DPS typically requires 1000 NFEs to achieve optimal results. In contrast, $\Pi$GDM serves as a competitive baseline and a strong foundation for our method. Additionally, $\Pi$GDM offers greater flexibility since it does not require a differentiable image degradation transform, unlike DPS. Thus, PGDM is a natural choice for us to demonstrate the effectiveness of our method.
>
> 2. Secondly, in the context of popular diffusions and flows, using the posterior approximation in $\Pi$GDM results in a closed-form derivation of the conjugate operator and its inverse (See Eqns. 11 and 12 in the main text). Note that this is not a limitation of our method since the projection operator can also be derived for DPS, as highlighted in the following theoretical result (for brevity, we omit the full proof and would include it in our revision).
>
> **Proposition.** For a time-dependent design matrix $B: [0,1] \rightarrow \mathbb{R}^{d\times d}$ and the posterior approximation proposed in DPS [Chung et al.] with guidance step size $\rho$ such that the conditional score $\nabla_{x_t} \log p(y|x_t) = \rho \frac{\partial \hat{x}_0}{\rho x_t}^\top H^\top (y - H\hat{x}_0)$, introducing the transformation $\hat{x}_t=A_tx_t$, where
> $$A_t = \exp{\Big[\int_0^t B_s - \Big(F_s + \frac{\rho}{2\mu_s^2}G_s G_s^\top H^\top H\Big)ds \Big]}$$ induces the following projected diffusion dynamics
> $$d\hat{x}_t = A_t B_t A_t^{-1}\hat{x}_t dt + d \Phi_y y + d \Phi_s \epsilon(x_t, t) + d \Phi_j \Big[\partial\epsilon(x_t, t)H^\top (y - H\hat{x}_0)\Big] $$
> where $\exp(.)$ denotes the matrix exponential, $\hat{x}_0$ represents the Tweedie's moment estimate and $H$ denotes the degradation operator. The proof roughly follows similar ideas from Appendix A.3 in our paper and we will include a complete proof in our revision.
>
> Despite our method applying to DPS, it is worth noting that in contrast to $\Pi$GDM, the projection operator $A_t$  and its inverse for DPS do not exhibit a closed-form solution and need to be approximated using perturbation analysis (if the step size $\rho$ is small) or computed using standard routines in packages like PyTorch. Therefore, though our method is also applicable to DPS, we choose to stick with the $\Pi$GDM framework in this work.
>
> > Can the method be applied to nonlinear inverse problems? If not, please clarify the reason.
>
> While our presentation of the proposed method is primarily in the context of linear-noiseless inverse problems, we also present an extension of our method to noisy-linear and non-linear inverse problems in Section 3.4. Furthermore, in Figure 11 of our paper, we illustrate the applicability of our method to non-linear inverse problems in the context of challenging problems like JPEG and lossy neural compression-based restoration. We would also like to point the reviewer to our common response (Point 1) for a more comprehensive justification of the utility of our method for noisy and non-linear inverse problems.
>
> > Though the Conditional Conjugate Integrators work well and can speed up the sampling process to 5 steps, a more detailed computational comparison is needed as each iteration becomes more complicated. Can you provide more detailed computational results?
>
> We provide a computational comparison in Table 1 using NFE as the metric, which is standard in the existing literature on inverse problems like DPS [Chung et al.] and PiGDM [Song et al.]. It is important to note that the compared models utilize the same pre-trained backbone diffusion/flow models, ensuring no differences in model inference speed. The only additional computation incurred involves a simple linear transform $A_t$ and some **scalar** integral calculations, which are computationally inexpensive. Moreover, these integrals can be precomputed offline (i.e., before sampling starts, as only the noise schedule influences the integral results). Therefore, the computational cost can be amortized between samples, reducing potential overhead. We will make this point more explicit in our revised manuscript.
>
> > Can you clarify the reason why you restrict yourself to the posterior approximation in $\Pi$GDM
>
> Please see our response to your first comment.

---

> > ### Comment · Reviewer_PpUR · 2024-08-12
> >
> > I appreciate the authors' efforts in preparing the rebuttal and acknowledge the theoretical contributions of the paper, which have been well-received by the other reviewers. However, I still have some concerns. The paper does not report PSNR and SSIM results, making it difficult to fully assess the quality of the proposed methods. Given that the paper utilizes the $\Pi$GDM paradigm, it is crucial to provide these metrics for a comprehensive evaluation. DPS, for example, demonstrates robustness in both perceptual and recovery metrics across various tasks, highlighting the importance of such evaluations. Considering above, I keep the original rating.

---

> ### Author Response · Authors · 2024-08-13
> **Author Response**
>
> We thank the reviewer for their response. Please find our detailed response below:
>
> **Comparison of perceptual vs Recovery metrics:** We agree that highlighting robustness in both perceptual and recovery metrics is important in the context of inverse problems. Following previous works like DPS and $\Pi$-GDM, we include only perceptual metrics in the main text since recovery metrics like PSNR and SSIM usually favor blurrier samples over perceptual quality. So, a tradeoff exists between distortion (a.k.a recovery)  and perceptual quality. However, whether perceptual quality/recovery is preferred depends on the application. Therefore, for completeness, we provide a comparison between DPS, $\Pi$GDM, and C-$\Pi$GDM (our method) in terms of PSNR, SSIM, FID, and LPIPS in Tables 1 and 2 on the ImageNet-256 and FFHQ-256 datasets on the 4x super-resolution task. It is worth noting that the PSNR and SSIM scores for all methods correspond with the best FID/LPIPS scores presented in the main text for these methods.
>
> Tables 1 and 2 show that our method achieves competitive PSNR and SSIM scores for better perceptual quality than competing methods, even for very small sampling budgets. For instance, on the FFHQ dataset, our method achieves a PSNR of 28.97 compared to 28.49 for DPS while achieving better perceptual sample quality (LPIPS: 0.095 for ours vs 0.107 for DPS) and requiring around 200 times less sampling budget (NFE=5 for our method vs 1000 for DPS). Therefore, we argue that our perceptual quality to recovery trade-off is much better than DPS, given our method is significantly faster than DPS.
>
> **Traversing the Recovery vs Perceptual trade-off in C-$\Pi$GDM**: In addition to the guidance weight $w$, our method also allows tuning an additional hyperparameter $\lambda$, which controls the dynamics of the projection operator (See Sections 2.3 and 3.2 for more intuition). Therefore, tuning $w$ and $\lambda$ can help traverse the trade-off curve between perceptual quality and distortion for a fixed NFE budget. We illustrate this aspect in Table 3 (fixed $\lambda$ with varying $w$) and Table 4 (fixed $w$ with varying $\lambda$) for the SR(x4) task on the ImageNet-256 dataset using the PSNR, LPIPS, and FID metrics. Therefore, our method offers greater flexibility to tune the sampling process towards either good perceptual quality or good recovery for a given application while maintaining the same number of sampling steps. In contrast, other methods like DPS do not offer such flexibility. Moreover, tuning the guidance weight in DPS is very expensive in the first place due to its high sampling budget requirement (around 1000 NFE).
>
> We will extend these comparisons for other tasks and add them in the Appendix section of our revised paper. We would also request the reviewer to reconsider their evaluation.
>
> |                     | PSNR  | SSIM  | FID   | LPIPS |
> |---------------------|-------|-------|-------|-------|
> | DPS (NFE=1000)      | 23.81 | 0.708 | 38.18 | 0.252 |
> | $\Pi$GDM (NFE=20)   | 21.92 | 0.646 | 37.36 | 0.222 |
> | C-$\Pi$GDM (NFE=5)  | 22.32 | 0.641 | 37.31 | 0.220 |
> | C-$\Pi$GDM (NFE=10) | 23.00 | 0.651 | 34.22 | 0.206 |
> | C-$\Pi$GDM (NFE=20) | 23.16 | 0.654 | 34.28 | 0.207 |
>
> **Table 1**: Comparison between different methods on ImageNet-256 for the SR(x4) task
>
> |                     | PSNR  | SSIM  | FID   | LPIPS |
> |---------------------|-------|-------|-------|-------|
> | DPS (NFE=1000)      | 28.49 | 0.834 | 30.86 | 0.107 |
> | $\Pi$GDM (NFE=20)   | 28.26 | 0.818 | 26.17 | 0.087 |
> | C-$\Pi$GDM (NFE=5)  | 28.97 | 0.832 | 32.01 | 0.095 |
> | C-$\Pi$GDM (NFE=10) | 29.03 | 0.821 | 29.07 | 0.086 |
> | C-$\Pi$GDM (NFE=20) | 28.79 | 0.809 | 26.37 | 0.083 |
>
> **Table 2**: Comparison between different methods on FFHQ-256 for the SR(x4) task.
>
> | w  | PSNR  | LPIPS | FID   |
> |----|-------|-------|-------|
> | 2  | 22.91 | 0.339 | 48.48 |
> | 4  | 23.37 | 0.306 | 45.03 |
> | 6  | 23.49 | 0.274 | 42.68 |
> | 8  | 23.44 | 0.266 | 40.96 |
> | 10 | 23.28 | 0.254 | 40.27 |
> | 12 | 22.89 | 0.246 | 40.13 |
> | 14 | 22.74 | 0.239 | 40.16 |
>
> **Table 3**: Illustration of the impact of $w$ for a fixed $\lambda=0.0$ on the sample recovery (PSNR) vs sample perceptual quality (LPIPS, FID) at NFE=5 for our method. The task is SR(x4) on the ImageNet-256 dataset.
>
> | $\lambda$ | PSNR  | LPIPS | FID   |
> |-----------|-------|-------|-------|
> | -1.0      | 20.96 | 0.291 | 42.56 |
> | -0.8      | 21.33 | 0.265 | 40.97 |
> | -0.6      | 21.69 | 0.240 | 39.38 |
> | -0.4      | 22.04 | 0.223 | 37.83 |
> | -0.2      | 22.32 | 0.220 | 37.31 |
> | 0.2       | 22.73 | 0.257 | 45.27 |
> | 0.4       | 22.90 | 0.275 | 48.98 |
> | 0.6       | 23.03 | 0.283 | 47.47 |
> | 0.8       | 23.11 | 0.285 | 46.2  |
> | 1.0       | 23.15 | 0.285 | 46.41 |
>
> **Table 4**: Illustration of the impact of $\lambda$ for a fixed $w=15$ on the sample recovery (PSNR) vs sample perceptual quality (LPIPS, FID) at NFE=5 for our method. The task is SR(x4) on the ImageNet-256 dataset.

---

### Official Review · Reviewer_hLds · 2024-07-12

**Soundness:** 3
**Presentation:** 3
**Contribution:** 3
**Rating:** 7
**Confidence:** 4

**Summary:**

This work proposes a plug-and-play based method that leverages pretrained diffusion and flow models to solve inverse problems. The proposed method called Conditional Conjugate Integrators adapts previously proposed Conjugate Integrators framework for fast sampling of diffusion and flow models to solve linear inverse problems. The key idea is to project the diffusion (flow) dynamics into another latent space where sampling can be more efficient. Upon completion, the dynamics is projected back to the original pixel space. The paper provides the mathematical forms of the projection operator for conditional diffusion (flow) dynamics, then adapts it to get the projection operator for linear inverse problems. The paper also provides tractable forms of the projection matrix and its inverse. This derivation can be seen as a more general form of previously proposed method $\Pi$GDM ($Pi$GFM). The method can be extended to nonlinear and noisy settings. The paper provides promising results on datasets like LSUN Bedrooms, AFHQ Cats, FFHQ, ImageNet etc. on tasks like inpainting, superresolution, gaussian deblurring etc.

**Strengths:**

1. Methodology: The proposed method seems efficient — it can solve linear inverse problems in 5 steps. In comparison, previous methods need 20-100 or even 1000 steps.
2. Experiments: The paper includes ablation studies on the choice of hyper-parameters and provides comparisons against former state-of-the-art methods like $\Pi$GDM, DPS, DDRM,$\Pi$GFM etc.  The proposed method out-performs previous methods by a significant margin while using 5-10 steps and is on par with or better than previous method for >=20 steps.
3. Writing: Paper is well written. The core ideas and the methodology have been presented well, and derivations are easy to follow.

**Weaknesses:**

The benefits of the proposed method for the settings of noisy linear inverse problem setting as well as non-linear inverse problems remain unclear. The paper does not include any quantitative results for these two problem settings.  It only provides some limited qualitative results for super-resolution for the noisy setting with $\sigma_y$=0.05 and compression inverse problem and JPEG restoration problem for non-linear settings.

**Questions:**

Questions:
1. Why does Eq. 117 hold true for VPSDE? I’m missing the simplification step here.
2. Why does Table 1 not include results of inpainting for diffusion models on ImageNet? Similarly, Table 4 in the appendix skips results for inpainting on FFHQ.

Suggestions:
1. All the tables and figures must explicitly state all the relevant settings of the inverse problems. It is not immediately apparent that some of these quantitative results are only for noiseless linear inverse problems.
2. Consider including an algorithmic box that summarizes C-$\Pi$GFM and  C-$\Pi$GDM. This would provide a concise overview of the method to the readers.

**Limitations:**

The paper discusses relevant limitations.

---

> ### Author Rebuttal · Authors · 2024-08-06
>
> $\newcommand\dag\dagger$
> We thank the reviewer for their insightful comments and feedback. We provide a detailed response to several concerns below:
>
> >The benefits of the proposed method for the settings of noisy linear inverse problem setting as well as non-linear inverse problems remain unclear. The paper does not include any quantitative results for these two problem settings. It only provides some limited qualitative results for super-resolution for the noisy setting with
> =0.05 and compression inverse problem and JPEG restoration problem for non-linear settings.
>
> As illustrated in Figures 7 and 11 in the main text, we would like to highlight that our proposed method can be applied to challenging noisy and non-linear inverse problems in as few as 5-10 sampling steps. Therefore, the proposed method can also be beneficial for these settings. However, we acknowledge that our qualitative results for noisy and nonlinear problems serve as a proof of concept and that further evaluation and refinements are necessary. We intend to pursue more detailed investigations and improvements in future work and will highlight the same in the Conclusion section of a subsequent revision. We would also like to point the reviewer to our common response (Point 1) for a more comprehensive justification of the utility of our method for noisy and non-linear inverse problems.
>
> >Why does Eq. 117 hold true for VPSDE? I’m missing the simplification step here.
>
> For a given degradation operator $H$, the core idea in this step is the property of the projection matrix $P=H^\top(HH^\top)^{-1}H$, which shows:
> $$PH^\dag = \big[H^\top(HH^\top)^{-1}H\big]\big[H^\top(HH^\top)^{-1}\big] = H^\top(HH^\top)^{-1} = H^\dag$$
> Now, Eqn. 116 reads as:
> $$\Phi_y = -\int_0^t \frac{w\beta_t\mu_s}{2}\exp(\kappa_1(s))\Big[H^\dag + (\exp(\kappa_2(s)) - 1)PH^\dag\Big] ds$$
> Replacing $PH^\dag = H^\dag$ in the above equation, we get,
>
> \begin{align}
> \Phi_y &= -\int_0^t \frac{w\beta_t\mu_s}{2}\exp(\kappa_1(s))\Big[H^\dag + (\exp(\kappa_2(s)) - 1)H^\dag\Big] ds\\\\
> &= -\int_0^t \frac{w\beta_t\mu_s}{2}\exp(\kappa_1(s))\exp(\kappa_2(s))H^\dag\Big] ds \\\\
> &= -\big[\int_0^t \frac{w\beta_t\mu_s}{2}\exp(\kappa_1(s) + \kappa_2(s))ds\big]H^\dag
> \end{align}
> Therefore, we arrive at Eqn. 117. We hope this resolves any confusion. We will also add these simplifying steps in our subsequent revision.
>
> > Why does Table 1 not include results of inpainting for diffusion models on ImageNet? Similarly, Table 4 in the appendix skips results for inpainting on FFHQ.
>
> We thank the reviewer for pointing this out. We couldn't include these comparisons due to time constraints but will include them in a subsequent revision.
>
> > All the tables and figures must explicitly state all the relevant settings of the inverse problems. It is not immediately apparent that some of these quantitative results are only for noiseless linear inverse problems.
>
> We thank the reviewer for pointing this out. We are planning to update the paper with more detailed captions and agree that we should mention the settings of the problems more explicitly in the figures and tables
>
> > Consider including an algorithmic box that summarizes C-$\Pi$GFM and C-$\Pi$GDM. This would provide a concise overview of the method to the readers.
>
> We thank the reviewer for this suggestion. We plan to add a pseudocode/algorithm box summarizing the proposed samplers in the main text. We would also like to point the reviewer to our common response (Point 2) for more details regarding improved illustrations of our method.

---

> > ### Comment · Reviewer_hLds · 2024-08-11
> >
> > I thank the authors for their response. I would like to retain my score.

---

### Author Rebuttal · Authors · 2024-08-06

We thank the reviewers for their insightful comments and are glad that the reviewers found our paper well-written (Reviewers hLds, PpUR), well-supported by theoretical arguments (Reviewer PpUR), our proposed method efficient (Reviewer hLds), and rigorously defined (Reviewer X2Z6). Below we highlight some shared concerns that the reviewers have and address reviewer-specific concerns as individual responses. We also provide a rebuttal pdf with our response which illustrates the main figure we plan on including in our paper revision for better understanding of the proposed method.

**Shared comments**:

**Re. applicability to Noisy and Nonlinear Inverse Problems**:

Reviewers hLds and PpUR wished to see more details and results of our approach applied to noisy and nonlinear inverse problems. While we appreciate the suggestions and are willing to expand on them (see below), we stress that our primary focus in this work is on the *linear and noiseless case*, which captures a set of diverse and important inverse problems encountered in image restoration applications (e.g., super-resolution, deblurring, and inpainting). Similar to other works (e.g., on $\Pi$GDM), we presented the selected results on nonlinear and noisy setups as proof of concept.

In our paper, we discuss our proposed approach for addressing noisy and nonlinear inverse problems in Section 3.4, with some qualitative results presented in Figures 7 and 11. More specifically, for noisy linear inverse problems, our approximations for the noisy projection operator $A_t^{\sigma_y}$ are accurate to an order of $O(\sigma_y^4)$ [See Eq. 15 in the main text], where $\sigma_y$ is the noise level added to the output of the degradation operator. Therefore, unless the noise levels are very high, this implies that our method is applicable in most practical scenarios. Empirically, Figure 7 illustrates the validity of our theoretical arguments for the task of noisy super-resolution. Therefore, while we do not include quantitative comparisons on noisy inverse problems, our qualitative results serve as a good proof-of-concept of the generality of our proposed conditional sampling framework. We will add further results for noisy-linear inverse problems in our revised manuscript to highlight this aspect further.

Next, for nonlinear problems, our approximations follow a heuristic approach that is similar to $\Pi$GDM (Song et al.) but can be empirically effective as demonstrated in the context of challenging non-linear inverse problems like JPEG and lossy neural compression-based restoration (see Figure 11) under a limited compute budget. Thus, our method offers promising avenues for speeding-up nonlinear inverse problems. However, we acknowledge that our qualitative results for noisy and nonlinear problems serve as a proof-of-concept, and that further evaluation and refinements are necessary. We intend to pursue more detailed investigations and improvements in future work.

**Re. Illustration of the proposed method**

Reviewers X2Z6 and hLds made nice suggestions about improving the illustration of our proposed method. As a follow-up, we present an intuitive visualization of our proposed sampling framework in Figure 1 in our rebuttal pdf. To summarize, as illustrated in the figure, given a starting time $t_s$, our method tries to map the conditional diffusion dynamics into a more well-conditioned space where sampling is more efficient. The hyperparameter \lambda controls the amenability of the projected space for faster sampling. Moreover, the projection operator itself is a function of the degradation operator and the guidance scale resulting in more robust sampling even at high guidance scales. We revert back to the original space after sampling concludes in the projected space. While we will continue to improve upon these visualizations, we hope that the accompanying figure can help resolve any confusion that the readers or the reviewers may have regarding our proposed approach. In subsequent revision, we will also include the algorithmic pseudocode of our method as suggested by Reviewer hLds for more clarity.

---

### Decision · Program_Chairs · 2024-09-25

**Decision:**

Accept (poster)

**Comment:**

This paper presents a numerical sampling method for solving inverse problems using diffusion and flow models. The proposed technique is based on conjugate integrators which provide high-quality solutions to linear inverse problems in a few sampling steps. Overall the reviewers have acknowledged the efficacy of the method and its rigorous treatment of the problem and they have recommended accepting the paper.